JCB Journal of Cell Biology

# Syncrip/hnRNP Q is required for activity-induced Msp300/Nesprin-1 expression and new synapse formation

Joshua Titlow[1], Francesca Robertson[1], Aino Järvelin[1], David Ish-Horowicz[1,2], Carlas Smith[3], Enrico Gratton[4], and Ilan Davis[1]

**Memory and learning involve activity-driven expression of proteins and cytoskeletal reorganization at new synapses, requiring posttranscriptional regulation of localized mRNA a long distance from corresponding nuclei. A key factor expressed early in synapse formation is Msp300/Nesprin-1, which organizes actin filaments around the new synapse. How Msp300 expression is regulated during synaptic plasticity is poorly understood. Here, we show that activity-dependent accumulation of Msp300 in the postsynaptic compartment of the *Drosophila* larval neuromuscular junction is regulated by the conserved RNA binding protein Syncrip/hnRNP Q. Syncrip (Syp) binds to *msp300* transcripts and is essential for plasticity. Single-molecule imaging shows that *msp300* is associated with Syp in vivo and forms ribosome-rich granules that contain the translation factor eIF4E. Elevated neural activity alters the dynamics of Syp and the number of *msp300*:Syp:eIF4E RNP granules at the synapse, suggesting that these particles facilitate translation. These results introduce Syp as an important early acting activity-dependent regulator of a plasticity gene that is strongly associated with human ataxias.**

## Introduction

Activity-dependent neuronal plasticity is the cellular basis of memory and learning, involving the formation of new synapses and cytoskeletal remodeling in response to neuronal activity (West and Greenberg, 2011). To achieve plasticity, it is thought that neuronal activation leads to the elevated expression of >1,000 different genes. Many activity-dependent genes have been identified through either RNA sequencing studies (Chen et al., 2016) or proteomics analysis (Dieterich and Kreutz, 2016), and the majority of the effort in the field has focused on explaining how altered neural activity leads to changes in gene expression through transcriptional regulation (Madabhushi and Kim, 2018). However, activity-dependent plasticity often occurs too rapidly and too far away from the cell nucleus to be explained by de novo transcription alone. Therefore, posttranscriptional regulation is thought to be a crucial mechanism to explain changes in gene expression in response to neuronal activity.

During synaptic plasticity, the actin cytoskeleton is extensively remodeled, a process requiring numerous regulatory proteins (Spence and Soderling, 2015). Nesprins are an especially interesting class of actin regulatory proteins because they connect both synapses and nuclei through the cytoskeleton. The Nesprins are encoded by genes called *synaptic nuclear envelope-1* and *-2* (*SYNE-1* and *-2*), which contain ≥80 disease-related variants that cause cerebellar ataxias or muscular dystrophies (Zhou et al., 2018b). The molecular function of Nesprins and their role in muscular diseases are relatively well studied in mouse models of *SYNE-1* and *SYNE-2* (Zhou et al., 2018a) in relation to nucleocytoplasmic and cytoskeletal organization and function, but the function of Nesprins in neurological disorders is not yet well understood.

The synaptic function of Nesprins has begun to be investigated in the *Drosophila melanogaster* orthologue, *msp300,* one of many molecular components that are conserved between the *Drosophila* larval neuromuscular junction (NMJ) and mammalian glutamatergic synapses (Harris and Littleton, 2015; Menon et al., 2013; Titlow and Cooper, 2018). Msp300 is required for activity-dependent plasticity at the larval NMJ (Packard et al., 2015), where it organizes a postsynaptic actin scaffold around newly formed synapse clusters, known as boutons. The postsynaptic actin scaffold regulates glutamate receptor density at the synapse (Blunk et al., 2014), which also requires Msp300 (Morel et al., 2014). Msp300 is barely detectable at mature NMJ synapses but becomes highly enriched at the postsynapse in response to elevated neural activity (Packard et al., 2015). However, the mechanism by which activity-dependent Msp300 expression is regulated is unknown.

[1]Department of Biochemistry, University of Oxford, Oxford, UK; [2]Medical Research Council Lab for Molecular Cell Biology, University College London, London, UK; [3]Centre for Neural Circuits and Behaviour, University of Oxford, Oxford, UK; [4]Laboratory for Fluorescence Dynamics, University of California Irvine, Irvine, CA.

Correspondence to Ilan Davis: ilan.davis@bioch.ox.ac.uk.

We previously identified *msp300* by RNA immunoprecipitation sequencing as the strongest interactor with an RNA binding protein (RBP) called Syncrip (Syp; McDermott et al., 2014). The mammalian orthologue of Syp is hnRNP Q (heterogeneous nuclear ribonuclear protein Q), an RBP that functions in a number of diverse biological processes ranging from sorting microRNA in exosome vesicles (Santangelo et al., 2016) to controlling the myeloid leukemia stem cell program (Vu et al., 2017) and was recently identified in a patient whole-exome sequencing study as a potential gene candidate for intellectual disability (Lelieveld et al., 2016). Syp is expressed throughout the mammalian brain (Tratnjek et al., 2017) and has been found in RNP particles with FMRP protein (Chen et al., 2012), IP3 mRNA (Bannai et al., 2004), and BC200 mRNA (Duning et al., 2008). Knockdown of Syp in rat cortical neurons throughout development increases neurite complexity and alters the localization of proteins encoded by its mRNA targets (Chen et al., 2012). Syp has also been shown to regulate the stability of its mRNA targets in macrophages (Kuchler et al., 2014). In the *Drosophila* larval NMJ, Syp is expressed postsynaptically, where it acts as a negative regulator of synapse development (Halstead et al., 2014; McDermott et al., 2014). Syp is required to maintain the correct synaptic pool of glutamatergic vesicles and therefore glutamatergic transmission at the larval NMJ. However, it is not known whether Syp regulates synapse formation or Msp300 expression in the context of activity-dependent synaptic plasticity.

Here, we show that Syp is required directly for synaptic plasticity and for regulating activity-induced Msp300 expression in larval muscles. Syp and *msp300* mRNA physically interact in vivo near the synapse in granules containing ribosomes and eukaryotic translation factor 4E (eIF4E), which become significantly less dynamic in response to elevated synaptic activity. Our work reveals a new RBP regulator that links neuronal activity to posttranscriptional control of an mRNA encoding an actin-binding protein that is essential for new synapse formation.

## Results

### Baseline and activity-dependent expression of Msp300 are posttranscriptionally regulated by Syp

In response to elevated neuronal activity, Msp300 is rapidly enriched at the larval NMJ (Fig. S1, A–C) where it is required for structural synaptic plasticity (Packard et al., 2015). We have previously shown that *msp300* mRNA is associated with Syp protein in immunoprecipitation experiments using whole larval lysates (McDermott et al., 2014). To determine whether Msp300 expression is regulated by Syp at the larval NMJ, we quantified *msp300* mRNA and protein levels in *syp* mutant fillet preparations and compared them to wild-type controls. We found that Msp300 expression at the larval NMJ is regulated posttranscriptionally by Syp. Primary nuclear *msp300* transcripts at the site of transcription (nascent transcripts) and cytosolic mRNA molecules were quantified using single-molecule FISH (smFISH; see Fig. S2 for details on probe design and controls). We used a previously characterized *syp* null allele, *syp^{e00286}* (McDermott et al., 2014), and found that the spatial distribution

of *msp300* transcripts in *syp* mutant muscles was indistinguishable from wild-type muscles (Fig. 1 A). The number of primary transcripts (relative intensity of transcription foci) and the number of mature transcripts were also not significantly altered in *syp^{e00286}* relative to wild-type larvae (Fig. 1 B). Therefore, loss of Syp does not affect mRNA localization or mRNA turnover. We also determined whether Msp300 protein levels are affected by loss of Syp, by quantifying Msp300 protein expression in larval muscles using immunofluorescence. Msp300 protein levels were on average 40% lower in *syp^{e00286}* than in wild-type controls (Fig. 1, C and D), which is in agreement with Western blot data from whole larvae showing that Syp is a positive regulator of Msp300 translation during NMJ development (McDermott et al., 2014). Taken together with our smFISH data, these results suggest that proper translation of *msp300* transcripts in larval muscle requires Syp, but *msp300* transcription and cytoplasmic mRNA levels are not regulated by Syp.

To determine whether Syp is required for the activity-dependent increase in Msp300, we measured Msp300 protein and mRNA levels in *syp* mutants and wild-type larvae in response to patterned KCl stimulation, a well-characterized model for synaptic plasticity that induces several of the physiological responses associated with behavioral, electrical, and optogenetic stimulated synapse growth at the larval NMJ (Fig. S1, D–H; Ataman et al., 2008; Sigrist et al., 2003). We found that Syp is required for postsynaptic enrichment of Msp300 in response to elevated synaptic activity. Msp300 protein levels in the muscle increased by ~40% in stimulated NMJs relative to mock-treated controls (Fig. 2, A–C). To determine if Syp is involved in elevating Msp300 levels, the experiment was repeated with conditional *syp* knockdown using the tripartite Gal80^{ts}/Gal4/upstream activating sequence (UAS) system (Suster et al., 2004; see Materials and methods for experimental details and Fig. S3 for RNAi controls). Conditional knockdown allowed us to separate potential developmental requirements for Syp from specific activity-dependent effects. We found that the conditional knockdown of *syp* almost completely abolished activity-dependent increases of Msp300 protein levels observed in wild-type muscles (Fig. 2, D and E). *msp300* mRNA levels were not significantly affected by neural activity (Fig. 2, F and G), indicating that Syp does not act through mRNA transcription or turnover. We conclude that Syp is required directly for elevating Msp300 protein levels in response to increased neural activity levels at the NMJ.

### Syp is required for activity-dependent plasticity at mature NMJ synapses

Syp was previously shown to be required for the correct development of synapse structure and synaptic transmission at the *Drosophila* larval NMJ (Halstead et al., 2014). To test whether Syp is also required for activity-dependent synaptic plasticity, we performed stimulus-induced plasticity assays in larval NMJs in which *syp* is either knocked out or conditionally knocked down. Our results reveal a specific requirement for Syp in activity-dependent plasticity of mature synapses, in addition to synapse development. To assess activity-dependent plasticity at the NMJ, we quantified new bouton formation and synaptic vesicle

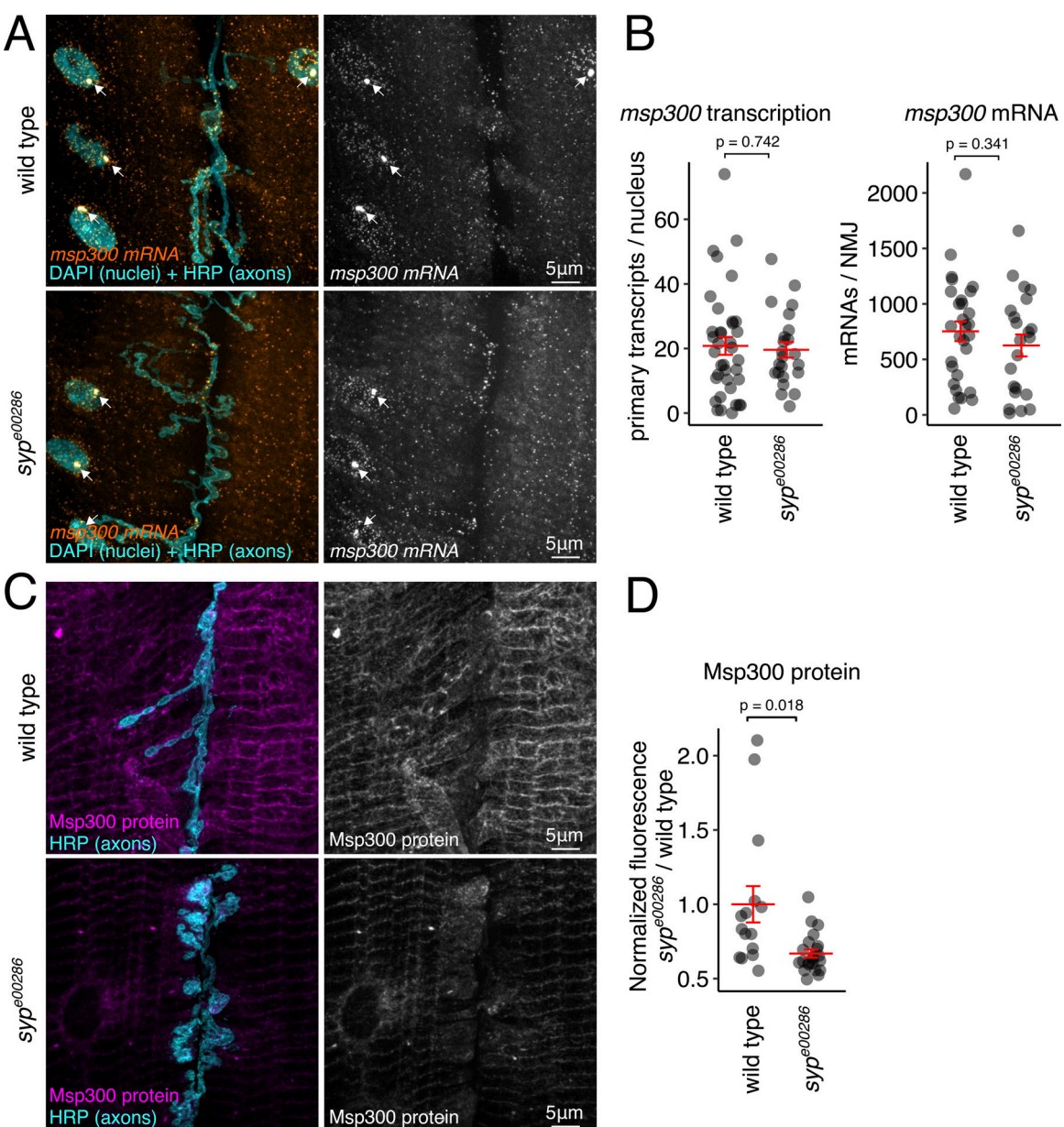

**Figure 1.** **Msp300 expression in larval muscle is regulated posttranscriptionally by Syp. (A)** *msp300* transcription and mRNA turnover are unaffected by loss of Syp. smFISH images show that steady-state levels of *msp300* transcription (arrows) and cytosolic mRNA at wild-type and *syp^{e00286}* mutant NMJs are similar. **(B)** Quantification shows that loss of *syp* does not have a significant effect on the level of primary or mature *msp300* transcripts at the larval NMJ. **(C)** Syp modulates Msp300 protein levels in larval muscle. Maximum z-projections of immunofluorescence images show that Msp300 protein levels in the muscles of *syp^{e00286}* mutant are reduced relative to wild-type larvae. **(D)** Quantification of Msp300 immunofluorescence shows that Msp300 protein levels are significantly reduced in *syp^{e00286}* relative to wild type. Mean ± SEM; Student's unpaired *t* test; number of NMJs measured shown in each bar.

release in Syp mutants after KCl stimulation (Ataman et al., 2008). The presence of immature boutons (ghost boutons [GBs]) was quantified by counting the number of boutons labeled by HRP immunofluorescence that also lack the postsynaptic density marker Dlg1 (arrows, Fig. 3 A). Wild-type larvae produced an average of 5 GBs per NMJ in response to KCl, while mock-stimulated controls had an average of 0.5 GBs per NMJ. In a *syp* null mutant, *syp^{e00286}*, the number of GBs induced by spaced KCl stimulation was twofold less than in wild-type larvae (Fig. 3 B). This phenotype is indeed due to loss of *syp* activity, because there was also a significant reduction in GBs in hemizygous

*syp^{e00286}*/*Df* NMJs, but not in a P-element excision revertant larvae (Fig. 3 B). Surprisingly, the few GBs that formed in the absence of Syp showed wild-type levels of Msp300 enrichment (Fig. S1, I–P). We conclude that Syp is required for structural plasticity at the larval NMJ.

To assess whether Syp also has a role in functional activity-dependent synaptic plasticity, we recorded miniature excitatory junction potentials (mEJPs) after spaced KCl stimulation. Stimulus-induced potentiation of mEJP frequency provides a physiological readout of NMJ plasticity (Ataman et al., 2008). In the Syp rescue line, mEJP frequency doubled in response to KCl

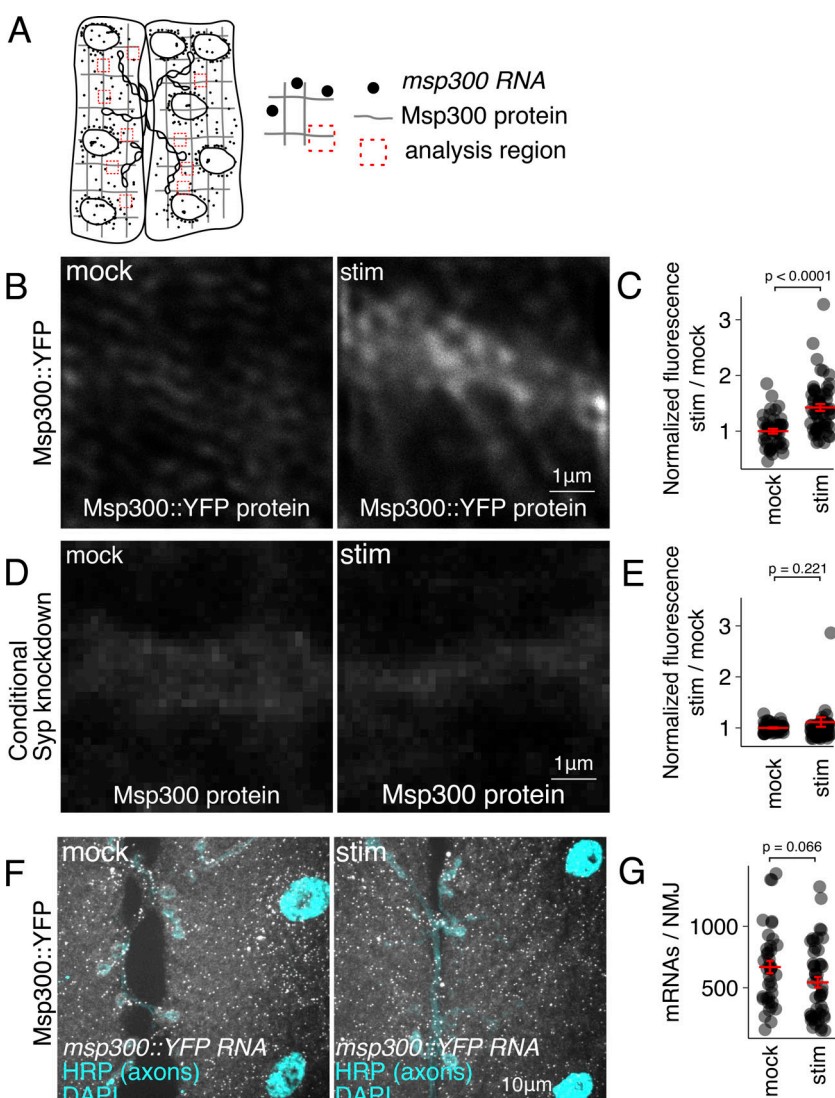

Figure 2. **Activity-induced Msp300 expression requires Syp and does not require de novo transcription. (A)** Schematic representing distribution of *msp300* RNA and protein in larval muscles, and regions of Msp300 accumulation on cytoskeletal actin filaments near the synapse (dotted squares) that were analyzed for immunofluorescence quantification (median intensity of 10 regions for each NMJ). **(B and C)** Msp300 protein levels in stimulated (stim) larval NMJs are increased 40% relative to nonstimulated control NMJs. **(D and E)** Activity-dependent increase in Msp300 protein level is inhibited when *syp* expression is knocked down for 24 h before the experiment. **(F and G)** Spaced potassium stimulation does not affect *msp300* mRNA levels at the NMJ. Mean ± SEM; Student's unpaired *t* test; number of NMJs measured shown in each bar.

stimulation but was significantly less elevated in *syp*[e00286] (Fig. 3 C), indicating that Syp is involved in activity-induced potentiation of synaptic vesicle release. Syp is likely to modulate vesicle release probability (and not quantal content), as mEJP amplitude was not affected by stimulus or genotype. Together, these results establish Syp as an important factor in structural and functional activity-dependent synaptic plasticity.

To separate the acute effects of Syp in synaptic plasticity from its developmental role in synapse formation, we again used conditional knockdown of *syp* with Gal80[ts]/Gal4/UAS. By isolating Syp's role at the mature NMJ from developmental effects, we were able to show that Syp acts immediately in response to neuronal activation to facilitate synaptic plasticity. Importantly, the NMJ morphology in conditional *syp* knockdown mutants was indistinguishable from wild-type NMJs (Fig. 3 D), indicating that synapse development was normal, in contrast to the NMJ developmental axon overgrowth phenotype observed in *syp*[e00286] (Fig. 1 C; McDermott et al., 2014). We found that while conditional *syp* overexpression did not interfere with GB formation in response to spaced KCl stimulation (Fig. 3 E), conditional *syp* knockdown completely inhibited KCl-induced GB formation

(Fig. 3 E). These experiments demonstrate a late larval stage requirement for Syp in activity-dependent plasticity at mature NMJ synapses.

## Genetic, biochemical, imaging, and biophysical evidence for interactions between Syp and *msp300* at the larval NMJ

Having found that Syp is required for activity-dependent Msp300 expression and that both proteins are required for synaptic plasticity, we performed a series of experiments to determine whether *syp* and *msp300* directly interact. First, we tested whether *syp* and *msp300* interact genetically. Homozygous *syp* and *msp300* mutants exhibit strong developmental phenotypes, as both mutants have structurally and functionally aberrant NMJs and the animals do not survive to the adult stage. Therefore, to test for genetic interactions in the context of normally developed larval NMJ synapses, we performed a transheterozygous genetic interaction experiment in *syp*[e00286]/*msp300*[Δ3'] larvae. Both are recessive alleles that show normal synapse development and activity-induced plasticity in heterozygous mutant larvae. However, activity-induced GB formation is completely abolished in the *syp*[e00286]/*msp300*[Δ3']

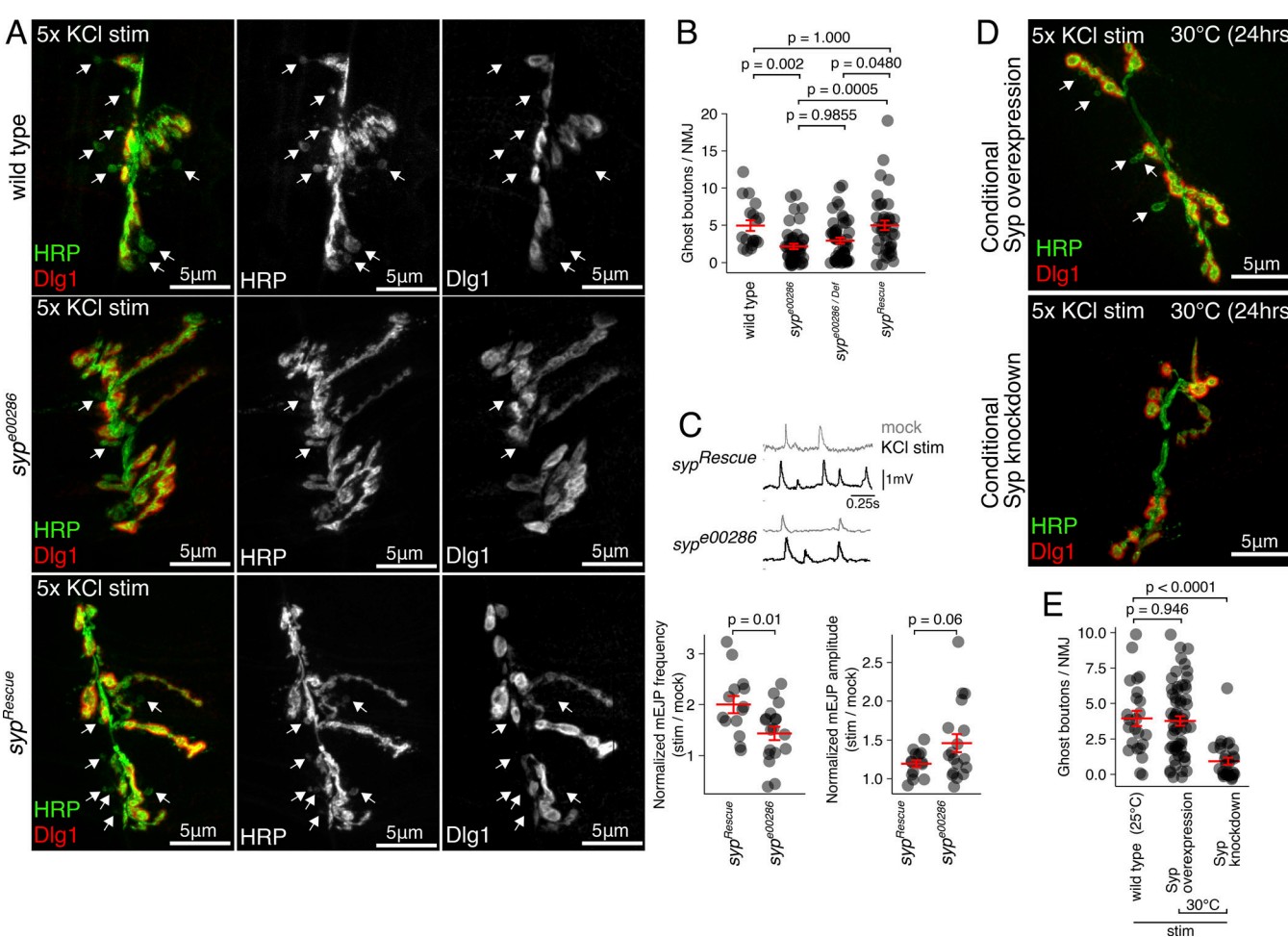

Figure 3. **Syp modulates activity-dependent synaptic plasticity in the larval NMJ, both developmentally and acutely, at mature synapses. (A)** New synaptic boutons (GBs; arrows) are formed by five rounds of KCl stimulation (stim) in wild-type NMJ preparations. GBs appear as immature HRP-positive axon terminals (green) that lack the postsynaptic density marker, Dlg1 (red). *syp* loss-of-function mutant (*syp^e00286*) has abnormal synapse morphology and relatively few stimulus-induced GBs. **(B)** Quantification of stimulus-induced GBs per NMJ comparing wild-type, *syp^e00286*, *syp^e00286*/Def, and P-element excision rescue larvae (mean ± SEM; Kruskal–Wallis with Dunn's post hoc test; number of NMJs shown in each bar). **(C)** Activity-induced potentiation of spontaneous synaptic vesicle release is inhibited in *syp* mutant larvae. Traces show mEJPs recorded from muscles after KCl stimulus or mock treatment in *syp* mutant and *syp* rescue lines. Histograms show the frequency and amplitude of mEJPs in stimulated muscles normalized to mock-treated larvae (mean ± SEM; Student's unpaired *t* test; number of muscles measured shown in each bar). **(D)** Larval NMJ morphology is unaffected by conditional overexpression or knockdown of *syp*. Representative maximum-projection confocal images of NMJs from KCl-stimulated larvae show the presence of GBs in the overexpression line, but not the conditional *syp* knockdown line. **(E)** Stimulus-induced GB formation is unaffected by conditional *syp* overexpression, but conditional knockdown with *syp* RNAi completely abolishes GB formation. Quantification of GB numbers (mean ± SEM; one-way ANOVA; number of NMJs is shown in each bar).

trans-heterozygous larvae (Fig. 4, A–E). This synthetic genetic interaction demonstrates that there is a functional link between *syp* and *msp300* that is specific to activity-dependent synaptic plasticity.

Next, we tested for biochemical interactions between Syp protein and *msp300* mRNA at the larval NMJ. Syp and *msp300* are both expressed postsynaptically at the NMJ, so we prepared lysates from dissected larval fillets after removing all other internal organs and the central nervous system. The presence of *msp300* mRNA in Syp immunoprecipitates was then quantified using reverse transcription quantitative PCR (qPCR). Enrichment of m*sp300* mRNA in the immunoprecipitation fractions was on average 50-fold higher than the nonbinding control transcript *rpl32* (Fig. 5 A), indicating that Syp associates with *msp300* transcripts in larval muscle, consistent with RNA

immunoprecipitation-qPCR experiments from whole larvae (McDermott et al., 2014).

For interactions between Syp and *msp300* to be functionally relevant to regulating activity-dependent plasticity, they should occur at or near synapses. Our experiments show that Syp granules near the synaptic boutons contain *msp300* mRNA. We used two different techniques to assess the interactions between *msp300* mRNA and Syp protein close to the NMJ. First, we visualized individual *msp300* transcripts and Syp-GFP fusion protein in fixed NMJs using superresolution confocal microscopy (Korobchevskaya et al., 2017). Syp was tagged with an N-terminal eGFP fusion protein at the endogenous locus (see Materials and methods for details). Expression of the Syp-GFP reporter is highly enriched in the nucleus and is also found in discrete punctae throughout the cytoplasm and near the

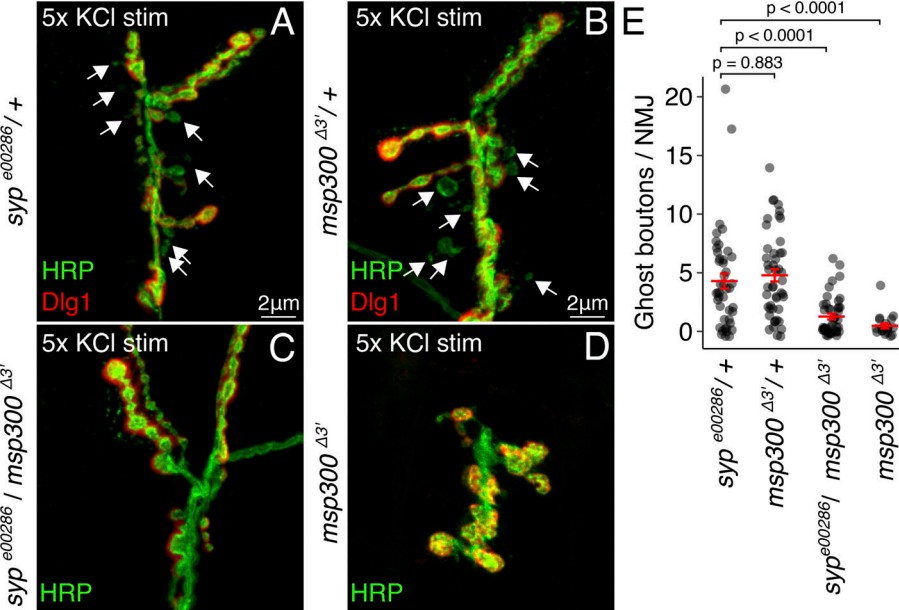

Figure 4. **_msp300_ and _syp_ show a strong genetic interaction in activity-dependent bouton formation. (A–D)** Trans-heterozygous _syp^{e00286}/msp300^{Δ3'}_ mutant NMJs have normal bouton morphology but fail to produce GBs in response to KCl stimulation (stim). Representative maximum-projection confocal images from KCl-stimulated NMJs show the presence of GBs in heterozygous _syp^{e00286}_ (A) and _dNesp1^{Δ3'}_ (B) mutants, but not the trans-heterozygous _syp^{e00286}/msp300^{Δ3'}_ mutants (C). **(D)** Homozygous _msp300^{Δ3'}_ mutants have significantly underdeveloped axon terminals and fail to produce GBs. **(E)** Quantification of KCl-induced GB formation shows that _syp^{e00286}/dNesp1^{Δ3'}_ mutants have significantly fewer activity-induced GBs than heterozygous controls (mean ± SEM; one-way ANOVA; number of NMJs is shown in each bar). Homozygous _msp300^{Δ3'}_ mutants and _syp^{e00286}_ (Fig. 1 B) also exhibit the inhibited GB phenotype.

synapse, as previously reported for Syp immunofluorescence (Fig. 5, B–D; Halstead et al., 2014). To covisualize Syp and _msp300_, we hybridized Syp-GFP larval fillet preparations with smFISH probes targeting _msp300_. We observed several _msp300_ mRNA molecules that spatially overlapped with Syp-GFP foci (Fig. 5, E–G, arrows), both in the cytoplasm and adjacent to the synapse. The presence of _msp300_ transcripts in Syp granules near the NMJ suggests that local translation of _msp300_ could be regulated by Syp.

To directly test for physical interactions between Syp protein and _msp300_ RNA in vivo, we covisualized _msp300_ RNA and Syp protein in live larval NMJ preparations using cross-correlation raster imaging correlation spectroscopy (ccRICS), a biophysical method that measures fluorescent protein complexes by virtue of correlated mobilities of individual molecules within an illuminated small rapidly scanned field (Digman et al., 2009). Our results showed significant interactions between Syp and _msp300_. To visualize _msp300_ RNA, we synthesized Cy5-tagged _msp300_ and microinjected it into the muscle cytoplasm of NMJ preparations from lines expressing endogenous Syp-GFP (Fig. 5, H–K). Molecular interaction between _msp300_ RNA and Syp-GFP was quantified by measuring the fraction of _msp300_ molecules that interact with Syp complexes. We found that on average, 31.0% ± 8% of _msp300_ molecules interact with Syp-GFP at the larval NMJ (nine cells from four different animals, _n_ = 22 recordings), compared with 0.08 ± 8% association with a control RNA and 3.1 ± 3% interaction with a control protein (described in more detail in the next paragraph). With ccRICS, we were also able to determine that the majority of these interactions between RNA and protein are dynamic, as opposed to stable, continuous associations (see Materials and methods for details).

We performed two negative controls to test whether the interaction between Syp-GFP and _msp300_ RNA is due to specific binding rather than a random nonspecific interaction. Both controls indicate that binding between Syp and _msp300_ is not an

artifact of the fluorophores or the binding assay itself. First, we repeated the ccRICS experiment with Syp-GFP and Cy5-labeled _rpl32_ RNA, which did not bind to Syp in our immunoprecipitation assay (Fig. 5 A). The fraction of _rpl32_ molecules bound to Syp-GFP was several orders of magnitude smaller than _msp300_ (0.08% ± 8%; _t_ test, P = 0.01; four experiments, _n_ = 15 cells from 10 animals, 30 different recordings). We also acquired ccRICS data from fluorescent _msp300_ RNA injected into muscles expressing free GFP and found that the fraction of _msp300_ molecules interacting with free GFP was significantly lower than the fraction interacting with Syp-GFP (−3.1 ± 3%, _t_ test, P = 0.001; one larva, three cells, four different recordings). From these experiments, we conclude that _msp300_ RNA dynamically interacts with Syp in RNP complexes at living NMJ synapses.

## Syp expression and protein dynamics are modulated by neuronal activity

Given Syp's role in mediating activity-dependent gene expression changes, we next asked how Syp granules respond to elevated neuronal activity. We tested whether neuronal activation alters Syp abundance or diffusion rate using RICS. Our results show that KCl stimulation causes an increase in Syp abundance and a significant decrease in its diffusion rate. Syp-GFP fluorescence was measured in living NMJ preparations that were either stimulated with KCl in HL3 saline buffer or mock treated with control HL3 saline. Images were acquired using the RICS format to measure Syp protein dynamics and Syp protein abundance from the same dataset (Fig. 6 A). We found that Syp protein expression was elevated 1.6-fold in the cytosol (P < 0.0001; Fig. 6, B and C) and 1.2-fold in the nucleus (P = 0.0274; Fig. 6, B and C) of KCl-stimulated samples relative to controls. From the same dataset, we used RICS analysis to determine the diffusion rate of Syp in vivo (Fig. 6, D and E). The average diffusion coefficient for Syp-GFP in mock-stimulated samples was 0.84 ± 0.04 μm²/s in the cytosol and 0.41 ± 0.03 μm²/s in the

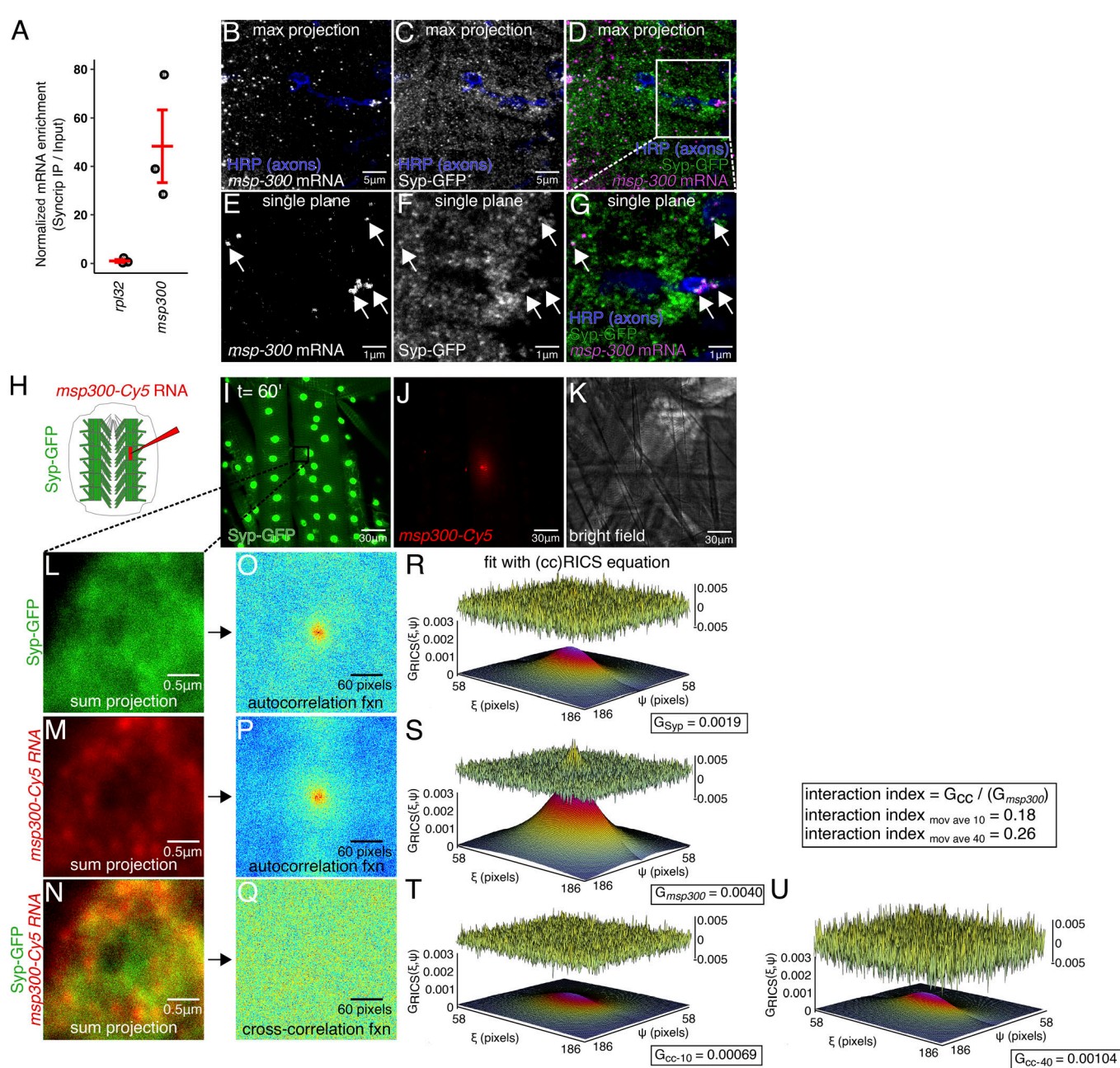

Figure 5. ***msp300* mRNA physically interacts with Syp granules near the larval NMJ, in vivo. (A)** *msp300* mRNA coprecipitates with Syp. Quantification of RT-qPCR data shows high enrichment of *msp300* relative to the nonbinding control *rpl32* (mean ± sem, *n* = 3 immunoprecipitations). **(B–D)** Representative confocal microscopy image of *msp300* smFISH and Syp-GFP signal at the larval NMJ, maximum z projection. **(E–G)** A single confocal slice of the same image, showing that *msp300* transcripts colocalize with Syp-containing RNP granules within the resolution limit of the system. **(H)** Schematic of Cy5-labeled *msp300* RNA injection experiment in larval NMJ preparation to test its association with Syp. **(I and J)** Representative live images of Syp-GFP in a larval muscle injected with Cy5-labeled *msp300* mRNA. **(K)** Bright-field image of an injected muscle indicates that the muscle still appears healthy 60 min after the injection. **(L–N)** Region of interest (from box in I) where ccRICS data were acquired. Each image is a sum projection of 50 images acquired in photon-counting mode. **(O–Q)** Autocorrelation function acquired from the images in L–N. **(R–U)** 3D plots of the spatial autocorrelation function with the relative amplitude and residuals after fitting with the RICS equation showing the relative fraction of molecular complexes containing both Syp protein and *msp300* mRNA. Fitting with an increased moving average reveals a higher proportion of dynamic interactions.

nucleus. In KCl-stimulated samples, there was a 25% decrease in Syp mobility relative to mock-treated samples, specifically in the cytosol near the NMJ synapses (Fig. 6 F), suggesting that the size of Syp complexes is altered in the stimulated state.

Activity-induced changes in the mobility of Syp could arise from a general increase in cytosolic viscosity. The activity-induced increase in Syp protein levels could arise from a general increase in translation. To determine if activity-induced changes in Syp mobility and translation are specific, and not a general nonspecific consequence of KCl stimulation on cellular viscosity and translation, we measured the effect of KCl stimulation on free GFP diffusion rate and protein expression levels

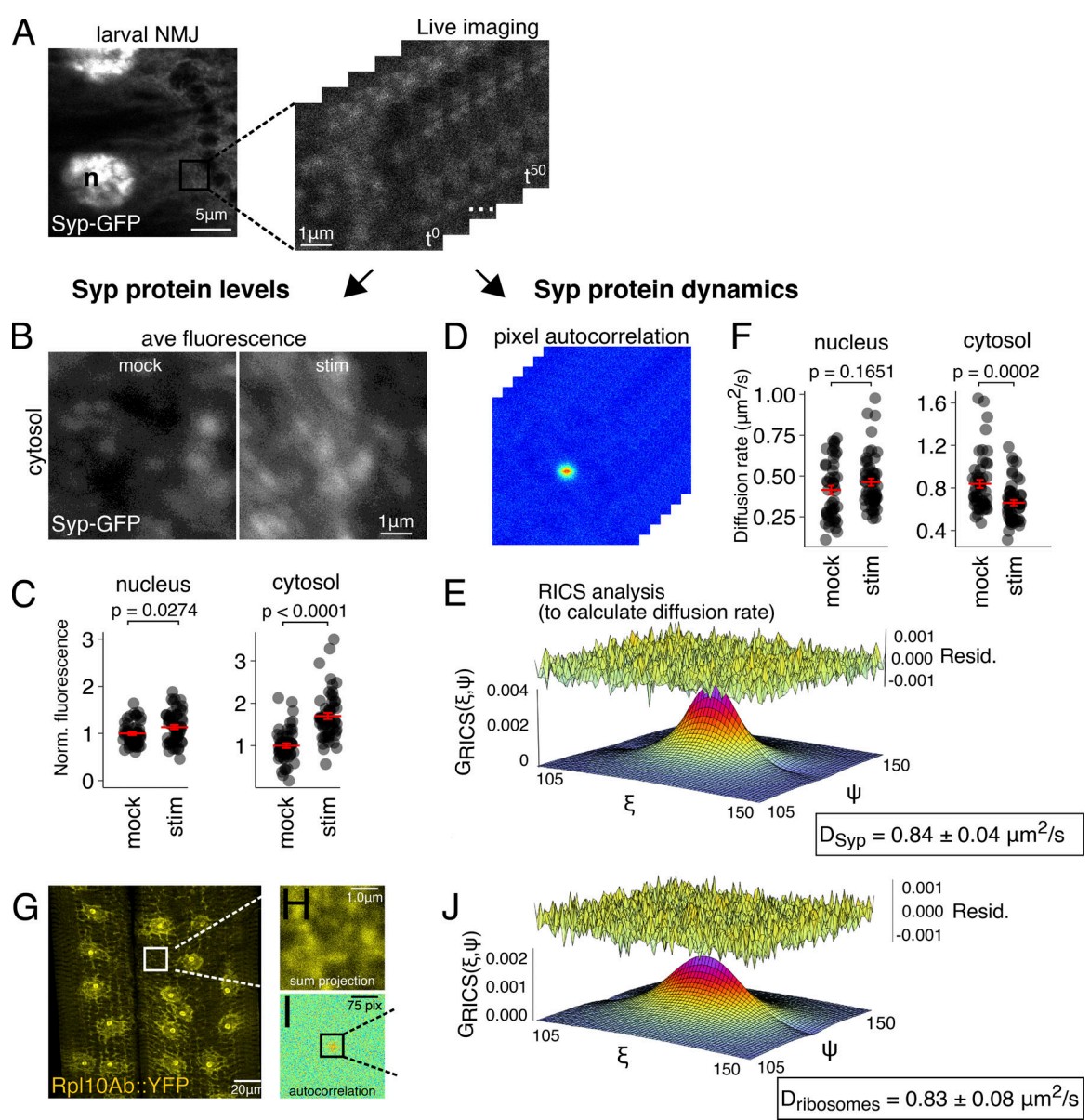

Figure 6.   **Synaptic activity modulates Syp protein levels and dynamics at the larval NMJ. (A)** A series of scanning confocal images were acquired in RICS format to measure both Syp protein levels and dynamics. **(B)** Average (ave)-intensity projection of an image time series shows that Syp-GFP levels at the NMJ are higher in KCl-stimulated samples relative to mock-stimulated controls. **(C)** Quantification of fluorescence intensity shows that Syp-GFP levels in KCl-stimulated samples are significantly higher in the cytosol, but not in the nucleus. **(D)** A plot of the spatial autocorrelation function for each image after averaging across the time series and subtracting the immobile fraction. **(E)** 3D plot of the autocorrelation function fitted with RICS. RICS analysis was used to calculate the apparent diffusion coefficient for Syp-GFP in the measured regions. **(F)** Syp-GFP diffusion rate was significantly reduced at the NMJ in KCl-stimulated samples relative to mock-stimulated controls. Nuclear Syp-GFP diffusion was unaffected. Mean ± SEM; Student's unpaired $t$ test; number of NMJs measured shown in each bar. **(G–J)** Diffusion rate of the large subunit ribosomal protein Rpl10A is almost identical to Syp. **(G)** Low-magnification image of Rpl10A::YFP shows that the tagged protein is properly localized in muscle, i.e., in the nucleolus and ER. Images taken in the RICS format (H) were used to calculate a spatial autocorrelation function (I) and determine the apparent diffusion coefficient (J).

(Fig. S4, A and B). In mock-treated larval muscles, the average GFP diffusion rate was 15.2 ± 4.7 μm²/s, which is similar to GFP diffusion rates measured with various techniques in mammalian cells (Gura Sadovsky et al., 2017). The GFP diffusion rate in larval muscles was not significantly altered by KCl stimulation (Fig. S4 C). Similarly, the level of cytosolic GFP was not significantly altered by KCl stimulation (Fig. S4 C). Therefore, our control experiments showed that elevated synaptic activity does not

cause a general, nonspecific increase in translation or a general change in the intracellular environment that influences protein diffusion. We conclude that activity-induced changes in Syp mobility and protein expression are a specific response to elevated synaptic activity.

To estimate the likely size of Syp-GFP complexes in living NMJs, we compared the measured diffusion rate of Syp-GFP to free GFP, as obtained in the measurements above. Based on size

alone (the Syp-GFP fusion protein is approximately threefold larger than GFP), we predict from the Stokes–Einstein equation that the diffusion rate of free GFP should be 1.4-fold faster than Syp-GFP (see Materials and methods for details of the calculations). The measured GFP diffusion rate in larval muscles was ~18-fold faster than Syp-GFP, which strongly suggests that Syp exists in a large molecular complex. In human cell culture, we also observed an unexpectedly slow Syp-GFP diffusion rate that was approximately eightfold slower than GFP (Fig. S4, E–I). These results are consistent with the notion that Syp is present in a large RNA granule in association with RBPs, and perhaps ribosomes.

To test whether the Syp complexes are likely to include ribosomes, we measured the diffusion rates of ribosomes in the larval NMJ. We found that ribosomal diffusion rates near the NMJ are very similar to Syp diffusion rates. Ribosome diffusion was measured by acquiring RICS data from a protein trap line with YFP inserted into the ribosomal Rpl10Ab gene (Lowe et al., 2014). The protein expression pattern of the Rpl10Ab::YFP protein trap line overlaps almost completely with an smFISH probe that detects 28s rRNA (Fig. S5 A), indicating that Rpl10Ab::YFP gets incorporated into ribosomes. RICS analysis revealed that the average diffusion rate of Rpl10Ab::YFP near the axon termini was ~0.83 ± 0.08 μm²/s (Fig. 6, G–J), which is nearly identical to that of the Syp-GFP diffusion rate. Furthermore, this diffusion rate is also similar to that measured for the large ribosomal subunit in mouse embryonic fibroblasts (Katz et al., 2016). These results suggest that Syp and ribosomes are both present at the larval NMJ in extremely large, similarly sized complexes.

### *msp300* mRNAs are localized at the larval NMJ in Syp mRNP granules with ribosomes and the translation factor eIF4E

Activity-dependent synapse formation at the larval NMJ requires translation (Ataman et al., 2008). To determine more directly whether *msp300* can be translated near the NMJ synapses, we visualized Syp mRNP granules and *msp300* together with ribosomes and the translation factor eIF4E. These experiments showed that a significant proportion of *msp300* molecules localize at the synapse within Syp granules that contain ribosomes and eIF4E. We first used superresolution microscopy to image Syp-GFP together with *28s rRNA* smFISH to establish whether they are in the same subregion of the cell. We found that Syp RNPs colocalize with *28s rRNA* and *msp300* mRNA, suggesting that ribosomes are present very near Syp RNP granules. Individual molecules of *28s rRNA* could not be resolved as individual puncta because ribosomes are present at such a high concentration in the larval muscle (Zhan et al., 2016).

To address this limitation, we tested multiple superresolution microscopy techniques, including Airyscan, 3D structured illumination microscopy (3D-SIM), and stimulated emission depletion (STED) microscopy. Although none of the techniques could resolve individual ribosomes, which measure less than ~30 nm in their longest axis (Scofield and Chooi, 1982; Verschoor et al., 1996, 1998), we achieved x,y resolution of 90–150 nm with the different techniques (Fig. S5, B–P). At this resolution, we observed discrete Syp-GFP particles overlapping

with *msp300* molecules and particles of *28s rRNA* in the subsynaptic reticulum that resemble clusters of ribosomes described in electron micrographs of the NMJ (Fig. 7, E–J; Ukken et al., 2016; Zhan et al., 2016). To quantify the specificity of colocalization, we determined the percentage of *msp300* molecules that occupy voxels with Syp-GFP granules, 28s rRNA, or both (see Fig. S5, Q–V; and Materials and methods for details). 31.5% ± 1.8% of *msp300* molecules localize within Syp granules, 38% ± 1.8% of *msp300* molecules occupy single voxels that have above-background 28s rRNA signal, and 11.2% ± 1.9% of *msp300* molecules colocalize with both Syp and *28s rRNA*. We assessed the statistical significance of the associations by comparing them to the percentage of *msp300* molecules that colocalize with the same number of randomly distributed pixels. Each of the biological associations was significantly higher than random associations (Student's unpaired *t* test, P < 0.0001; n = 674 molecules from 19 NMJs in six larvae). We conclude that Syp is very closely localized with ribosomes near the synapses, consistent with the idea that their coassociation with mRNA in granules that are translationally competent.

We next asked whether *msp300* is associated with the essential translation factor eIF4E, which is a rate-limiting component for assembly of the mRNA–ribosome complex (Rhoads, 1993). eIF4E is also known to accumulate with poly-A binding protein (another translation initiation factor) at the larval NMJ in response to elevated synaptic activity, where it is required for local translation of GluRIIA (Menon et al., 2004; Sigrist et al., 2000, 2002, 2003). We found that large synaptic eIF4E granules contain *msp300* RNA (Fig. 7, H–M) and that the percentage of *msp300*-containing eIF4E granules increases in response to KCl stimulation (Fig. 7, N–P). To visualize eIF4E at the larval NMJ, we used an eIF4E::GFP fluorescent reporter (protein trap in the endogenous locus, documented in the Materials and methods section) combined with smFISH to visualize *msp300* molecules. We found that the number of large eIF4E::GFP granules per NMJ doubles in response to KCl stimulation (Fig. 7 N) and that the percentage of NMJs that express *msp300*-containing eIF4E granules increases by greater than fivefold after stimulation (Fig. 7 O), with no significant change in the number of *msp300* molecules per eIF4E granule (Fig. 7 P). We interpret the localization of *msp300* with ribosomes and eIF4E near the NMJ synapses as an indication that Syp is able to rapidly facilitate translation at the synapse in response to elevated neural activity (Fig. 8).

## Discussion

In this study, we describe a posttranscriptional mechanism that is important for activity-induced gene expression and synaptic plasticity. Accumulation of the actin binding protein Msp300 (an orthologue of human Nesprin-1), which is required for new synapse formation, is regulated at the mRNA level by an RBP called Syp (an orthologue of human hnRNP Q). Genetic analyses indicate that *msp300* and *syp* are together required for new bouton formation at the *Drosophila* larval NMJ. We have shown that Syp is required specifically for synaptic plasticity independently of its longer-term role in the development of

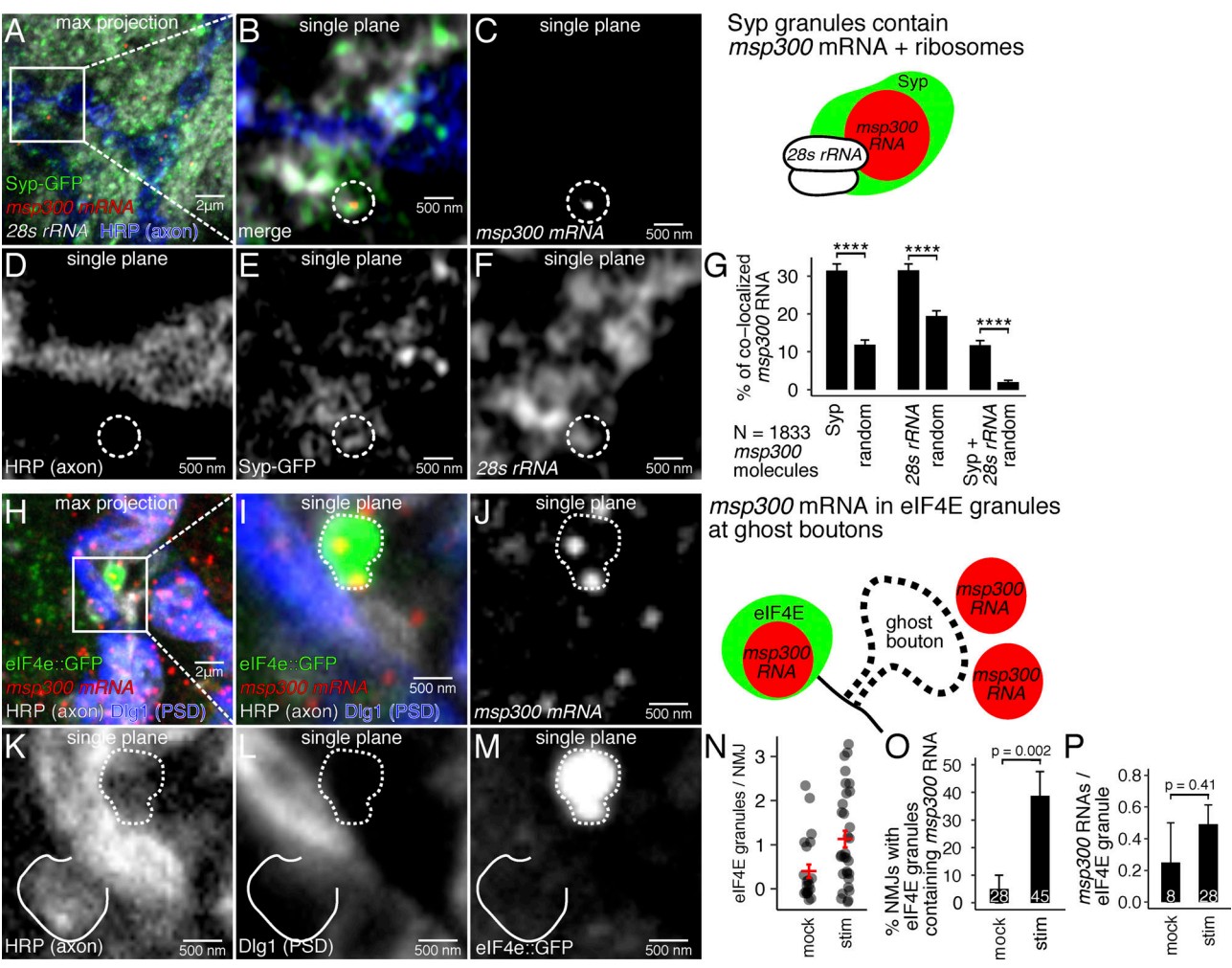

Figure 7. **msp300 mRNA is localized with Syp granules and translation machinery at the larval NMJ. (A–G)** Syp granules at the larval NMJ contain *msp300* mRNA and ribosomes. **(A)** Representative maximum z-projection of Airyscan superresolution images showing Syp-GFP (green), *msp300* mRNA (red), and *28s rRNA* (white) at the NMJ (blue). **(B–F)** Magnified single-plane images show an example of an *msp300* transcript and *28s rRNA* residing within a Syp granule (dotted circle). The percentage of *msp300* RNA molecules colocalizing with Syp, *28s rRNA*, or both was quantified and compared with randomly distributed signal (G; ****, *t* test, P < 0.0001; 674 *msp300* molecules from 18 different cells, 6 different animals). **(H–P)** *msp300* is present in large, activity-induced eIF4E granules (dotted outline) at GBs (solid line). **(N and O)** The number of large, synaptic eIF4E granules and the percentage of eIF4E granules containing *msp300* RNA significantly increases in KCl-stimulated NMJs (Student's unpaired *t* test; bars show mean ± SEM and the number of NMJs per condition). **(P)** The number of *msp300* RNA molecules per eIF4E granule is not affected by KCl stimulation (Student's unpaired *t* test; bars show mean ± SEM and the number of granules per condition).

structurally correct synapses. Using ccRICS and single-molecule imaging, we found that *msp300* mRNA is associated with Syp complexes in the postsynaptic compartment. Syp/*msp300* complexes also associate with ribosomes and the rate-limiting translation initiation factor eIF4E. Together, these data show that Syp regulates the formation of new boutons during activity-dependent synaptic plasticity through posttranscriptional control of msp300 expression.

The presence of ribosomes, eIF4E, and *msp300* RNA in Syp granules (Fig. 7) suggests that they are translationally competent mRNP granules. Moreover, the mobility of Syp granules changes in response to stimulation, which is consistent with an increase in ribosome density in the granule as a consequence of increased translation initiation. This idea is compatible with a previous report showing that Syp influences the level of protein produced

from *grk* in the *Drosophila* oocyte during egg chamber development (McDermott et al., 2012). Syp has also been shown to facilitate translation in a wide range of biological contexts, including HIV-1 Gag-p24 RNA (Vincendeau et al., 2013), Hepatitis C viral mRNA (Kim et al., 2004), the circadian clock gene Per1 (Lee et al., 2012), and the p53 tumor suppressor (Kim et al., 2013).

Based on our current findings and the previous literature on Syp in *Drosophila* and mammals, we propose a model in which Syp facilitates translation of *msp300* near the synapse, enabling extra accumulation of Msp300 that organizes actin filaments for local transport and maturation of the postsynaptic density (Fig. 8; Packard et al., 2015). This model is supported by colocalization and single-molecule biophysical interactions that provide strong evidence for local regulation of *msp300* mRNA at

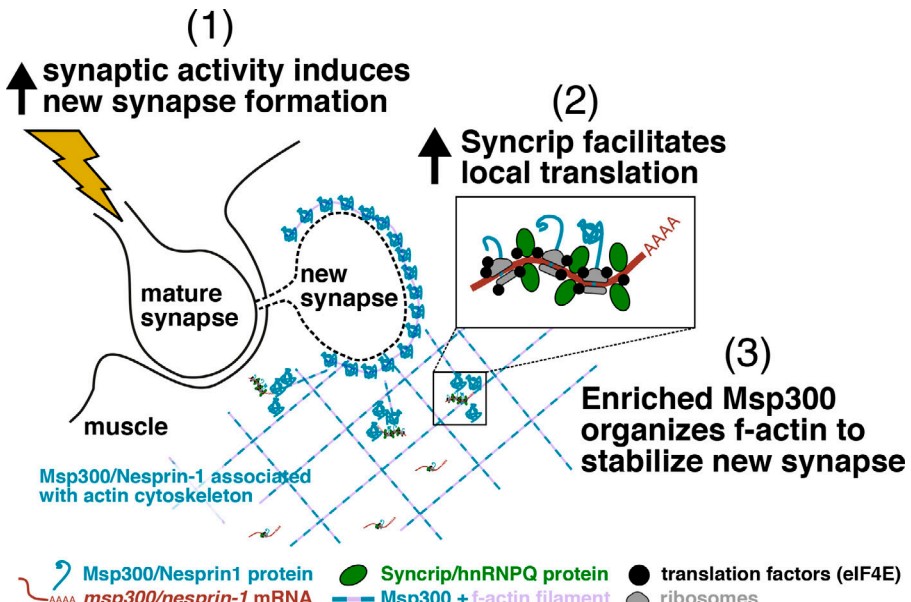

Figure 8. **Proposed mechanism for Syp's role in regulating activity-dependent enrichment of Msp300 and new synapse formation.** Synaptic activity at the larval NMJ is elevated by increased crawling behavior in the animal, which induces the formation of a new synaptic bouton. Msp300 is rapidly enriched around the new bouton to organize an actin scaffold where postsynaptic proteins will be anchored. *msp300* mRNAs at the synapse are present in a ribosome-containing complex with Syp and translation factors (e.g., eIF4E) that become much less dynamic in response to elevated activity. We hypothesize that Syp facilitates activity-dependent translation of *msp300* at the synapse by enabling recruitment of additional ribosomes and translation factors to the mRNA. This local pool of new Msp300 protein becomes enriched at the GB to stabilize the synapse by organizing an actin scaffold.

the synapse. Our data do not rule out the possibility that existing Msp300 protein is redistributed to new synapses upon activation or that new Msp300 is translated at a moderate distance from the synapse and transported there. We observe Msp300 at GBs in the absence of *syp* (Fig. S1, I–P); however, these alternative sources of Msp300 are not sufficient for producing normal levels of activity-dependent bouton growth (Fig. 3, B and E). It is also likely that Syp is required for the activity-dependent accumulation of other proteins required for synapse formation, as Syp is known to bind dozens of mRNAs coding for synaptic proteins (McDermott et al., 2014).

"Local" translation in neurons refers to protein synthesis that occurs within axons or dendrites, independent of the cell body, and usually in response to a specific stimulus that induces synaptic plasticity. The distance between the site of local translation and the functional site of the new proteins varies from 1 to 20 µm depending on cell type (Rangaraju et al., 2017). In the postsynaptic (muscle) compartment of the *Drosophila* larval NMJ, our definition of local translation refers to protein synthesis that occurs within 5 µm of mature axon terminals, which is separated by ≤30 µm from the majority of ribosomes found in rough ER around the muscle nuclei. The local region within 5 µm of mature axon terminals corresponds to the area in which most new boutons are established during plasticity and is the region in which we observe localized *msp300* mRNA associated with ribosomes and eIF4E, as well as an activity-induced increase in Msp300 levels that requires Syp.

Based on published eukaryotic translation and transcription rates, it is extremely unlikely that *msp300* could be transcribed, processed, exported, translated, and transported from the muscle nuclei to the synapse within the duration of our assay (150 min). The *msp300* gene and primary transcripts are 110 kb long, encoding 13,000 amino acids. We estimate that the gene would take 110 min to transcribe at 1 kb/min (Fukaya et al., 2017) and 65 min to translate at 3 amino acids/s (Riba et al., 2019). Therefore, we conclude that translation of existing *msp300* in

cytoplasm nearer to axon termini is necessary to supply the local pool of Msp300 protein that we observe accumulating at newly formed synapses.

*Drosophila msp300* is orthologous to the mammalian genes *SYNE-1* and *SYNE-2*. *msp300* and *SYNE-1/2* both encode Nesprin proteins that perform many similar functions, including regulation of glutamate receptor expression (Cottrell et al., 2004; Morel et al., 2014) and positioning of myonuclei (Stroud et al., 2017; Volk, 2013; Wang et al., 2015; Zhou et al., 2018a). The functional role of Nesprins in the nervous system is more enigmatic and requires further attention, because mutations in *SYNE-1* and -2 are strongly linked to recessive forms of hereditary cerebellar and extracerebellar ataxias in humans (Dupré et al., 2007; Gros-Louis et al., 2007; Noreau et al., 2013; Synofzik et al., 2016; Wiethoff et al., 2016). The molecular function that causes central nervous system–specific defects in these ataxias is not known, although it has been shown that neurogenesis and neuronal migration are significantly impaired (Zhang et al., 2009) and that white matter and cerebellar and cortical regions of the brain are significantly disrupted in patients (Gama et al., 2018). The *SYNE-1* ataxias also demonstrate extracerebellar phenotypes that are similar to neurodegenerative disease (Gama et al., 2016; Mademan et al., 2016). The most common mutations already observed to be linked with *SYNE-1* ataxias cause truncations or abnormal splice junctions. Our results raise the possibility that impaired posttranscriptional regulation of *SYNE-1* exacerbates the ataxia phenotypes.

Our work provides insight into how the actin cytoskeleton is remodeled during activity-dependent synaptic plasticity. A filamentous actin scaffold is required at new synapses to anchor postsynaptic density proteins, which in turn anchor postsynaptic receptors. Msp300 is one of the first proteins to assemble at newly formed synaptic boutons (Fig. S1), where it is thought to facilitate actin polymerization by recruiting an unconventional myosin from the Myosin ID family, Myo31DF in *Drosophila* (Packard et al., 2015). Myo31DF forms a complex with Arp2/3 to

mediate actin nucleation (Evangelista et al., 2000). How these actin-regulating factors assemble at the synapse is not yet known, but local translation is an attractive mechanism, since it is already well established that β-actin is translated from localized mRNA in mammalian dendrites (Buxbaum et al., 2014; Eom et al., 2003; Katz et al., 2016). There is also extensive evidence showing that *arc1* mRNA, a cytoskeleton-associated immediate early gene, localizes to dendrites and is locally translated (Guzowski et al., 2000; Steward et al., 1998; Steward and Worley, 2001). Several additional mRNAs that encode actin regulating proteins, including Arp2/3, have been identified by sequencing transcriptomes specifically from dendritic compartments (Will et al., 2013), and Syp binds several mRNAs in addition to *msp300* that encode actin-regulating proteins. Thus, future work should address whether posttranscriptional regulation of localized actin regulating proteins, by Syp in particular, is a conserved mechanism that is important for synaptic plasticity throughout the brain.

In conclusion, we identified a key component of posttranscriptional regulation during activity-dependent synaptic plasticity at the larval NMJ, which provides insight into how an actin binding protein is locally enriched to organize the postsynaptic scaffold for new synaptic boutons. Syp and Msp300 are known to be present at various synapses in other organisms, so it is tempting to speculate that interactions between Syp and *msp300* transcripts will be required for synaptic plasticity in those systems. Our study also lays the groundwork for studying biophysical properties of mRNA and associated granules at intact synapses, in vivo. Extending this approach to other molecules will provide a more complete picture of how the cell consolidates experience into new synapses.

## Materials and methods

### *Drosophila* maintenance
Fly stocks were maintained with standard cornmeal food at 25°C on 12-h light–dark cycles unless otherwise specified. Wandering third instar larvae were used for all experiments. The following genotypes were used: Oregon R (wild type unless otherwise specified), *syp^e00286* (McDermott et al., 2014), MHC-Gal4 muscle driver, UAS-Syp-GFP (McDermott et al., 2014), C57-Gal4 muscle driver, tubulin-Gal80^ts; and the following MS2/MS2 coat protein (MCP) lines: hsp83-MCP-mCherry (Hayashi et al., 2014), hsp83-MCP-GFP, and *grk*-MS2 × 12 (described in Jaramillo et al., 2008); Msp300::YFP (CPTI003472; Lowe et al., 2014); and eIF4E::GFP (Buszczak et al., 2007). The UAS-Syp RNAi line was obtained from Vienna Drosophila Stock Center and was previously characterized in the larval NMJ (McDermott et al., 2014). All other lines were obtained from Bloomington Drosophila Stock Center.

### Whole-mount smFISH and immunofluorescence
Stimulated larval NMJ specimens or mock-treated controls were prepared using a protocol that has been described previously (Titlow et al., 2018). Briefly, specimens were fixed in PFA (4% in PBS with 0.3% Triton X-100 [PBTX]) for 25 min, rinsed three times in PBTX, blocked for 30 min in PBTX + BSA (1%), and

incubated overnight at 37°C in hybe solution (2× SSC, 10% formamide, 10% dextran-sulfate, smFISH probes [250 nm; individual probe sequences listed in Table S1], and primary antibodies). The next morning, samples were rinsed three times in smFISH wash buffer (2× SSC + 10% formamide) and incubated for 45 min at 37°C in smFISH wash buffer with secondary antibodies and DAPI (1 µg/ml), and then washed for 30 min in smFISH wash buffer at room temperature before mounting in glycerol (Vectashield). PBTX was used in place of smFISH wash buffer for experiments that did not require smFISH. The following antibodies were used: mouse anti-Dlg1 (1:500; 4F3, Developmental Studies Hybridoma Bank), guinea pig anti-Syp (1:500; McDermott et al., 2012), guinea pig anti-Msp300 (1:1,000; Volk, 1992), HRP-Dyelight-405/488/Alexa Fluor 568/Alexa Fluor 659 (1:100; Jackson ImmunoResearch Laboratories), donkey anti-guinea pig Alexa Fluor 488 (1:500; Thermo Fisher Scientific), and donkey anti-mouse Alexa Fluor 568 (1:500; Thermo Fisher Scientific).

### Image acquisition and analysis
Whole-mounted immunofluorescence and smFISH specimens were imaged on a spinning disk confocal microscope (UltraView VoX; PerkinElmer) with 60× oil objective (1.35 NA, UPlan SApo, Olympus) and electron-multiplying charge-coupled device camera (ImagEM; Hamamatsu Photonics). NMJs at muscles 6 and 7 in segments 3–5 were imaged for at least five different larvae per condition/genotype and multiple cells per larvae unless specified otherwise. GBs were counted manually. Immunofluorescence signal intensity was quantified by measuring the median of 10 small regions (1 × 1-µm square) from average intensity z-projections of each NMJ. Mature and nascent transcripts were counted using a Matlab program called FISHquant (Mueller et al., 2013). Images in Fig. 2 B were deconvolved for display purposes using the Richardson–Lucy algorithm in the ImageJ (National Institutes of Health) DeconvolutionLab2 plugin (20 iterations, Airy PSF; Sage et al., 2017).

Superresolution images were acquired on an LSM-880 (Zeiss) with Airyscan detector and 60×/1.4-NA oil objective. Main pinhole was adjusted to 2.0 a.u. with a 0.2-a.u. pinhole in each of the 32 individual channels. Voxel size was set to 40 nm in x and y and 150 nm in z. To correct for chromatic aberration, we labeled the DNA with Vybrant DyeCycle Violet Stain (which emits from blue to far-red spectra) and acquired z-stacks in each emission channel with 405-nm excitation. Chromagnon (Matsuda et al., 2018) was then used to apply chromatic shift correction to images in which colocalization was assessed.

To assess the performance of Airyscan relative to other superresolution microscopy techniques, we acquired images of *28s rRNA* smFISH at the larval NMJ using SIM and STED microscopy. The x,y spatial resolution of each modality was then estimated from fast Fourier transform radial plots generated in SimCheck (Fig. S5, G–P; Ball et al., 2015). SIM images were acquired on a DeltaVision OMX V3 (GE Healthcare) with 60×/1.42-NA oil objective (PLAPON 60XO NA1.42; Olympus) and Cascade II 512 electron-multiplying charge-coupled device cameras (Photometrics). SIM reconstruction was performed with softWoRx (GE). STED images were acquired on a Leica TCS

SP8 STED 3× inverted microscope with HC PL APO 93×/1.30-NA glycerol objective and GaAsP HyD detector.

## Colocalization analysis

To quantify the colocalization of *msp300* molecules with a second reporter (Syp-GFP, *28s rRNA*, or eIF4E-GFP), we calculated the percentage of *msp300* molecules that occupied the same 3D voxel as the second reporter. An overview of the image analysis routine is as follows: acquire 3D confocal images for each channel, 3D chromatic shift correction, centroid analysis of *msp300*, background subtraction of the second reporter, threshold and banalization of the second reporter, Boolean assessment of colocalization between *msp300* and the second reporter, randomization of the second reporter signal, and then reassessment of colocalization. Chromatic shift correction was performed with Chromagnon, as described above. Centroid position of *msp300* molecules was determined by fitting a 3D Gaussian function of the smFISH point spread function in FISH quant (Mueller et al., 2013). Background subtraction was performed on the second reporter channel using rolling ball subtraction in FIJI (radius = 50 pixels; Schindelin et al., 2012, 2015), followed by autothresholding with either the Otsu or RenyiEntropy algorithm. We then used a custom Python script to map each *msp300* centroid position back to its associated voxel in the second reporter channel and determine whether the signal intensity in that voxel was above background, i.e., in the mask or not. Pixels in the second reporter channel were then randomly sorted with the random.shuffle() Python module, and the percentage of *msp300* molecules above threshold was recalculated to determine statistical significance of colocalization. Importantly, the number of pixels in both calculations was equal.

## Spaced potassium stimulation protocol

Six third instar larvae were dissected in two separate chambers to allow even saline perfusion from peristaltic pumps. A series of five short high potassium saline (KCl, 90 mM) pulses (2, 2, 2, 4, and 6 min) were separated by 15-min perfusion of HL3 saline as described previously (Ataman et al., 2008; McDermott et al., 2012). For smFISH and immunofluorescence, the larvae were fixed 150 min after the first stimulus. For electrophysiology and live imaging experiments, the recordings were made from 10 min after the last stimulus.

## Electrophysiology

Groups of three larvae were analyzed after chemical activation or mock treatments. Intracellular recordings were made in muscle 6 in segments 3–5 using sharp glass electrodes (10–20 MΩ) filled with 3 M KCl. Miniature excitatory postsynaptic potentials were amplified with a Multiclamp 200B, digitized with a Digidata 1550A A/D board controlled with pClamp (v10, Molecular Devices), and analyzed offline using Mini Analysis software (v6.0.3, Synaptosoft). Spontaneous activity was recorded for 2 min, and mEJP amplitude and frequency were analyzed for the second minute.

An extracellular nerve stimulation assay was used to acutely elevate motor neuron activity in the larval NMJ fillet preparation. A glass suction Ag/Cl electrode (~1 μm diameter) was filled with HL3 saline and attached to nerve roots of the posterior segments, leaving the segmental nerves and central nervous system fully intact. Superthreshold voltage pulses (0.02 ms × 0.75 V) were delivered through a stimulus isolation unit triggered by a Digidata 1550A A/D board controlled with pClamp. The stimulus amplitude was determined to be superthreshold both by visual monitoring of induced body wall contractions and by recording EJPs from the muscle fibers as described above. 10-pulse stimulus trains (40 Hz) were delivered every 5 s for 5 min, followed by 15-min recovery. The paradigm was repeated four times to provide a physiological comparison to the patterned KCl stimulation assay. After the stimulus, tissue was fixed and prepared for immunohistochemical detection of GBs as described above.

## RNA immunoprecipitation and RT-qPCR

For each biological replicate, 10 third instar larval body walls were dissected in HL3 medium and homogenized in immunoprecipitation buffer (50 mM Tris-HCl, pH 8.0, 150 mM NaCl, 0.5% NP-40, 10% glycerol, 1 mini tablet of Complete EDTA-free protease inhibitor [Roche], and RNAsin [Promega]). Lysates were incubated overnight at 4°C with magnetic Dynabeads (Thermo Fisher Scientific) conjugated to guinea pig anti-Syp and IgG antibody. Beads were washed four times briefly with cold lysis buffer. To retrieve the RNA, beads were resuspended in elution buffer (50 mM Tris-HCl, pH 8.0, 10 mM EDTA, 1.3% SDS, and RNAsin) and incubated at 65°C, 1,000 rpm for 30 min on a thermomixer. The elution step was repeated, and the supernatants were pooled. RNA was then extracted using an Illustra RNAspin mini kit (GE Healthcare). Input and eluate samples were used for cDNA synthesis using RevertAid Premium Reverse Transcription (Thermo Fisher Scientific). cDNA was used directly as a template for real-time PCR (SYBR green, Bio-Rad). Primer sequences were as follows: rpl32 forward, 5′-GCTAAG CTGTCGCACAAATG-3′, and reverse, 5′-TCCGGTGGGCAGCAT GTG-3′; msp-300 forward, 5′-TGCGCGATAAGGAGCAACAG-3′, and reverse, 5′-ATGAGGAGCTGTTCCGTTTGG-3′.

## Generation of Syp-GFP fly line

The syp-AttP line was generated using CRISPR to delete a 4-kb section at the beginning of the Syp coding region, which was replaced by an AttP site. sgRNA construct design and validation was performed by Dr. Andrew Basset, Genome Engineering Oxford. 1-kb homology arms, corresponding to sequences flanking the sgRNA cleavage sites (located in the 5′ UTR and third intron of isoform F), were cloned into the pDsRed-attP vector (Addgene, 51019). sgRNA constructs and the homology construct were injected in vas-cas9 embryos (BL 51323) by the Cambridge Fly facility. Embryos from the syp-AttP line were then injected with an AttB construct (RIV[Cherry]; Baena-Lopez et al., 2013) containing eGFP fused to the N terminus of Syp. sgRNA guide sites were as follows: 5′-TGCGTTCGTTGAACTCTA CAAGG-3′ and 5′-CCTTTCGATTTGGGGGGGATATGG-3′.

## Doxycycline-induced expression of GFP and Syp-GFP in HeLa cell lines

To generate stable cell lines, eGFP and human Syp-GFP plasmids were cloned into the Flp-In expression vector and integrated

into Flp-In 293 T-REx cells using standard procedures. For RICS imaging experiments, the cells were grown to 50% confluence in duplicate cultures on six-well plates (9 cm$^2$) and induced with doxycycline (0.10 µg/ml in clear DMEM with 10% BSA) for 6 h before imaging. Cells were then imaged in a temperature-controlled chamber at 37°C.

### RICS and ccRICS

The RICS method derives the apparent molecular diffusion rate from calculation of the spatial autocorrelation function between points in a scanning confocal image (Brown et al., 2008; Digman et al., 2009; Rossow et al., 2010). RICS data were acquired on a Zeiss LSM-880 upright confocal system using a 20×/1.0-NA water-immersion objective (Plan Apo; Olympus) and GaSP detector in photon counting mode. Laser power, pixel dwell time (8.19 µs), pixel size (20 nm), and frame size (256 × 256 pixels) were kept constant for all specimens, and mock-treated specimens were always measured in parallel with stimulated specimens. Fluorescence intensity was quantified as average raw pixel intensity values from an average of 50 individual frames.

Calibration data were acquired as described above for live imaging, i.e., 50 frames (256 × 256) were acquired with constant pixel width (20 nm), pixel dwell time (8.19 µs), and line scan time (4.92 ms). Calibration images were performed each day to determine the size of the beam waste. Donkey IgG conjugated to Alexa Fluor 488 (10 nM) was imaged with the settings described above, and the data were fitted in SimFCS software (Brown et al., 2008) using 40 µm$^2$/s as the defined value for diffusion coefficient (Arrio-Dupont et al., 2000). A moving average of 10 frames was applied to remove artifacts from cell movement, and the data were fitted with a single-component model, as residual plots and χ$^2$ values revealed an acceptable goodness of fit.

ccRICS was used to calculate the proportion of fluorescent *msp300* RNA molecules interacting with Syp-GFP complexes. Images of both molecules were acquired simultaneously from two channels, corrected for chromatic shift as described above (using 100-nm Tetraspek beads instead of Violet Dye), and fitted with RICS and ccRICS equations. The interaction index was calculated by dividing the amplitude of the ccRICS autocorrelation function ($G_{cc}$) by the amplitude of the RICS autocorrelation function of the *msp300* channel ($G_{msp300}$). The result is an interaction index ($G_{cc}/G_{msp300}$) that is proportional to the fraction of *msp300* molecules interacting with Syp complexes. We calculated the amplitude of the cross-correlation RICS function ($G_{cc}$) using a moving average of 10 frames to subtract the immobile fraction or a moving average of 40 frames to assess binding interactions (Digman et al., 2009). The interaction index ($G_{cc}/G_{msp300}$) derived from a ccRICS function with a moving average of 10 frames was 0.13 ± 0.04, which means that ~13% of *msp300* mRNA molecules diffuse in a complex with Syp molecules. The interaction index was 0.39 ± 0.10 when derived from a ccRICS function with a moving average of 40 frames, which means that ~39% of *msp300* mRNA molecules interact transiently with Syp complexes. As a negative control, we measured the interaction index for *msp300-Cy5* RNA injected into larval muscles expressing cytosolic GFP. We found that *msp300-Cy5*

was anti-correlated with cytosolic GFP, regardless of whether the moving average for the ccRICS function was 10 frames (interaction index = −0.002 ± 0.0005) or 40 frames (interaction index = −0.03 ± 0.03).

### Fluorescence correlation spectroscopy

Fluorescence correlation spectroscopy data were acquired on a Zeiss LSM-880 upright confocal system using a 20× 1.0-NA water-immersion objective and GaASP detector with factory photon counter. Photon counts were acquired at 15 MHz for 10-s intervals at different spots throughout the muscle, and the correlated data (1-ms bins) were saved as .fcs files. The autocorrelation function was fitted using open source software, FoCus-point, with the standard equation for 3D diffusion:

$$G_{3D}(t_c) = \sum_{k=1}^{D_s} \cdot A_k \left\{ \left[ 1 + \left( \frac{t_c}{\tau_{xy_k}} \right)^{\alpha_k} \right]^{-1} \left[ 1 + \left( \frac{t_c}{AR_k^2} \times \tau_{xy_k} \right) \right] \right\}^{-1/2},$$

where $G_{3D}$ is the amplitude of the correlation function, $t_c$ represents time, $D_s$ is the number of diffusing species, $A_k$ is a factor that establishes the proportion of each diffusing species, $\tau_{xy}$ is the lateral diffusion rate, $\alpha$ is the diffusion anomalous diffusion factor, and $AR$ is a constant factor that relates the axial diffusion rate to the lateral diffusion rate (Waithe et al., 2016). For GFP diffusion in the muscle, the autocorrelation was fitted with a one-species diffusion model, anomalous diffusion factor of 1, and $AR$ factor of 5.

### In vitro transcription and microinjection of Cy5-labeled RNA

Template cDNA from the *Drosophila* Gold collection (Msp300-HL01686, Rpl32-RE59709; Rubin et al., 2000) was amplified by bacterial transformation using a standard protocol from the Drosophila Genomics Resource Center. The plasmid DNA (5 µg) was linearized in an overnight digestion reaction with appropriate enzymes (Msp300-Apal and Rpl32-BamHI), and cleavage was verified by gel electrophoresis. After purifying the plasmid with QIAquick PCR Purification Kit, RNA was transcribed in a 50-µl reaction according to the polymerase manufacturer's instructions using the following components: DNA polymerase (Msp300-T3, Rpl32-T7; 20 units; Thermo Fisher Scientific) and associated transcription buffer, linear DNA (1 µg), mCAP analogue (Stratagene), DTT (1 M), rNTP mix-UTP (10 mM CTP, 10 mM ATP, and 3 mM GTP), Cy5-labeled UTP mix (1:1 mixture of labeled and unlabeled UTP, 10 mM total [UTP]), and RNase inhibitor (40 units; Promega). Template DNA was digested with RNase-free DNase1 (2.0 units; Qiagen), and Cy5-labeled RNA was purified using a Sephadex G50 spin column (Roche mini-Quick Spin RNA column), followed by EtOH precipitation. Purified RNA was then diluted to 100 ng/µl with RNase-free water for injection.

Cy5-labeled RNA was pressure injected into *Drosophila* larval muscles using prefabricated glass capillary tips (0.5-µm inner diameter, 1.0-µm outer diameter; Eppendorf Femtotips). Short pulses (3–5 × 100 ms) were delivered into muscle 6, and delivery of the labeled RNA was verified by epifluorescence. Specimens were then transferred to a Zeiss LSM-880 scanning confocal microscope for ccRICS analysis (described above).

### Statistical analysis of GB, mEJPs, mRNA, and protein levels

Statistical tests that were applied to each dataset are given in the figure legends along with the number of samples appearing in each graph. The normality assumption was tested with the Shapiro–Wilk test. The equal variances assumption was tested with an $F$ test or Levene's test, depending on the number of groups. Normally distributed populations with equal variances were compared using Student's $t$ test or one-way ANOVA (with Tukey test for multiple comparisons), depending on the number of groups. Populations with nonnormal distributions were compared using the Wilcoxon rank sum test or Kruskal–Wallis test (with Dunn test for multiple comparisons), depending on the number of groups. All statistical analyses were performed in R (v3.3.2 running in Jupyter Notebook).

### Online supplemental material

Fig. S1 shows a comparison of GB formation in response to KCl stimulation and motor nerve (electrical) stimulation (A–H). The figure also shows quantification of Msp300 protein enrichment at GBs (I–P). Fig. S2 shows where the smFISH probes and antibodies target *msp300* gene products, and negative and positive controls for each reagent. Fig. S3 shows quantification of *syp* knockdown in the larval NMJ. Fig. S4 shows control experiments for RICS in *Drosophila* larval NMJ and in HEK cells. Fig. S5 shows the lateral resolution of various imaging techniques for detecting 28s rRNA signal in the larval NMJ with smFISH (A–P). The figure also demonstrates how colocalization of *msp300* with Syp and 28s rRNA was quantified (Q–V). Table S1 lists the individual probe sequences used for smFISH.

## Acknowledgments

We are very grateful to the Bloomington Drosophila Stock Centre (fly stocks and cDNA) and to Flybase for their reagents and open data, which were both invaluable to this work. We thank members of the Davis laboratory for critical reading of the manuscript; Dr. Talila Volk (Weizmann Institute of Science, Rehovot, Israel) for antibodies, stocks, and discussions; Drs. Christoffer Lagerholm, Richard Parton, Lothar Schermelleh, and Pablo Hernandez-Varas for assistance and advice on super-resolution microscopy and specimen preparation; and three anonymous reviewers who provided excellent suggestions that significantly improved the final manuscript.

This work was supported by a Wellcome Trust Senior Research Fellowship (096144) and Wellcome Trust Investigator Award (209412) to I. Davis. Advanced microscopy facilities and technical advice were provided by Micron (https://micronoxford.com), supported by Wellcome Trust Strategic Awards (091911 and 107457) and a Medical Research Council/Engineering and Physical Sciences Research Council/Biotechnology and Biological Sciences Research Council next-generation imaging award to I. Davis as the principal applicant. D. Ish-Horowicz was supported by University College London. J. Titlow was supported by a Leverhulme Trust grant to I. Davis. E. Gratton was supported by the National Institute of General Medical Sciences, National Institutes of Health, grant 2P41GM103540. F. Robertson was funded by a Marie Skłodowska-Curie Postdoctoral Fellowship.

The authors declare no competing financial interests.

Author contributions: J. Titlow and I. Davis conceived and designed the study, interpreted the results, and wrote and revised the manuscript. D. Ish-Horowicz provided critical discussion of the results and revisions of the manuscript. J. Titlow performed the majority of the experiments, reagent generation, and data analysis. F. Robertson and A. Järvelin generated additional reagents and performed additional experiments. C. Smith and E. Gratton performed additional data analysis.

Submitted: 22 March 2019

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

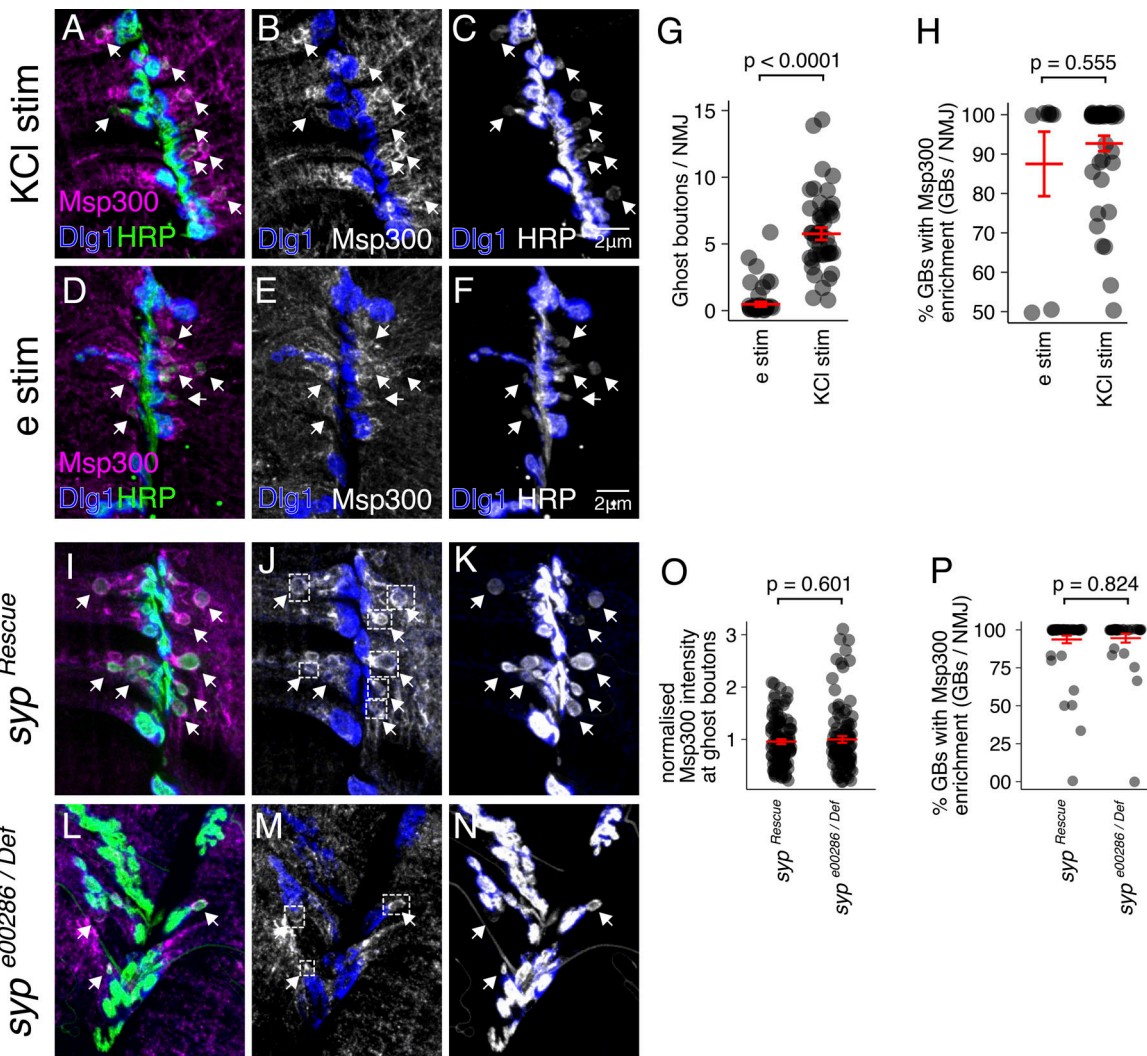

Figure S1. **Msp300 protein is enriched around activity-induced GBs. (A–C)** Maximum z-projection of spinning disk confocal images show strong Msp300 immunofluorescence around GBs (arrows) after spaced KCl stimulation (stim), but not around mature synapses (labeled by Dlg1, blue). **(D–F)** Msp300 protein also rapidly accumulates around GBs induced by electrophysiological stimulation (e stim). Electrophysiological stimulation induces fewer GBs per NMJ than KCl stimulation (G), but the percentage of GBs that show Msp300 enrichment is ∼90% for both stimuli (H; Student's unpaired *t* test; mean ± SEM). **(I–N)** To determine if Syp affects Msp300 accumulation at GBs, we compared the intensity of Msp300 accumulation and percentage of GBs showing Msp300 accumulation in the *syp*^Rescue^ and *syp*^e00286/Def^ genotypes. The intensity of Msp300 at GBs (O; representative regions of interest shown in white boxes in J and M) and the percentage of GBs with Mp300 enrichment (P) were unaffected by loss of Syp (Student's unpaired *t* test; mean ± SEM).

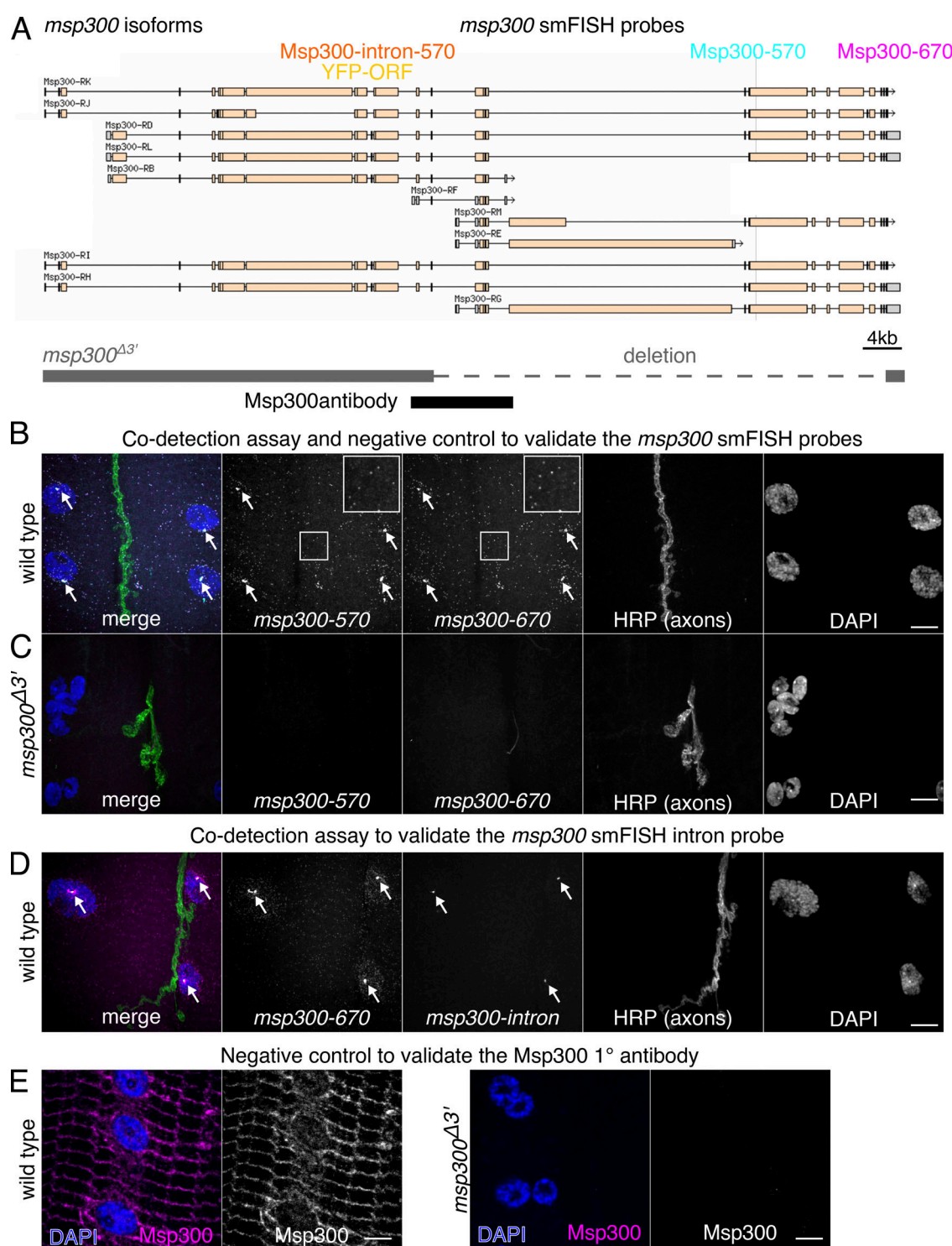

Figure S2. **smFISH probes for *msp300* have high detection efficiency and specificity. (A)** Schematic of *msp300* mRNA isoforms showing the position of smFISH probes and the YFP protein trap insertion used in this study. smFISH probes were designed to target large exons at the 3′ end of the transcript that are common to most isoforms (turquoise and magenta boxes), with dyes that have easily distinguishable fluorescence emission spectra, i.e., Quasar-570 (Msp300-570) and Quasar-670 (Msp300-670). **(B)** A codetection assay shows that signals from both *msp300* smFISH exon probes appear as bright punctae throughout the muscle cytoplasm with brighter transcription foci (arrows) in the larval NMJ. The cytosolic spots have a uniform intensity, and the majority of spots are detected in both channels (insets), indications that the signal arises from single molecules and that the detection efficiency is high. **(C)** smFISH probes for *msp300* do not show any off-target binding. Punctate smFISH signal is not observed in an Msp300 mutant (*msp300^Δ3′*) that was hybridized and imaged under identical acquisition settings as in B. **(D)** Intron/exon smFISH codetection assay shows that the large nuclear foci in B are primary *msp300* transcripts. Signal from an smFISH probe targeting intronic sequence overlaps with the Msp300-670 exon probe but does not label mature mRNA in the nucleus or cytoplasm. Images are maximum-intensity projections of spinning disk confocal sections; scale bars = 10 µm. **(E)** The Msp300 antibody specifically detects Msp300, as no signal is observed in the *msp300^Δ3′* mutant (note the nuclear aggregation phenotype that is indicative of *msp300* loss of function).

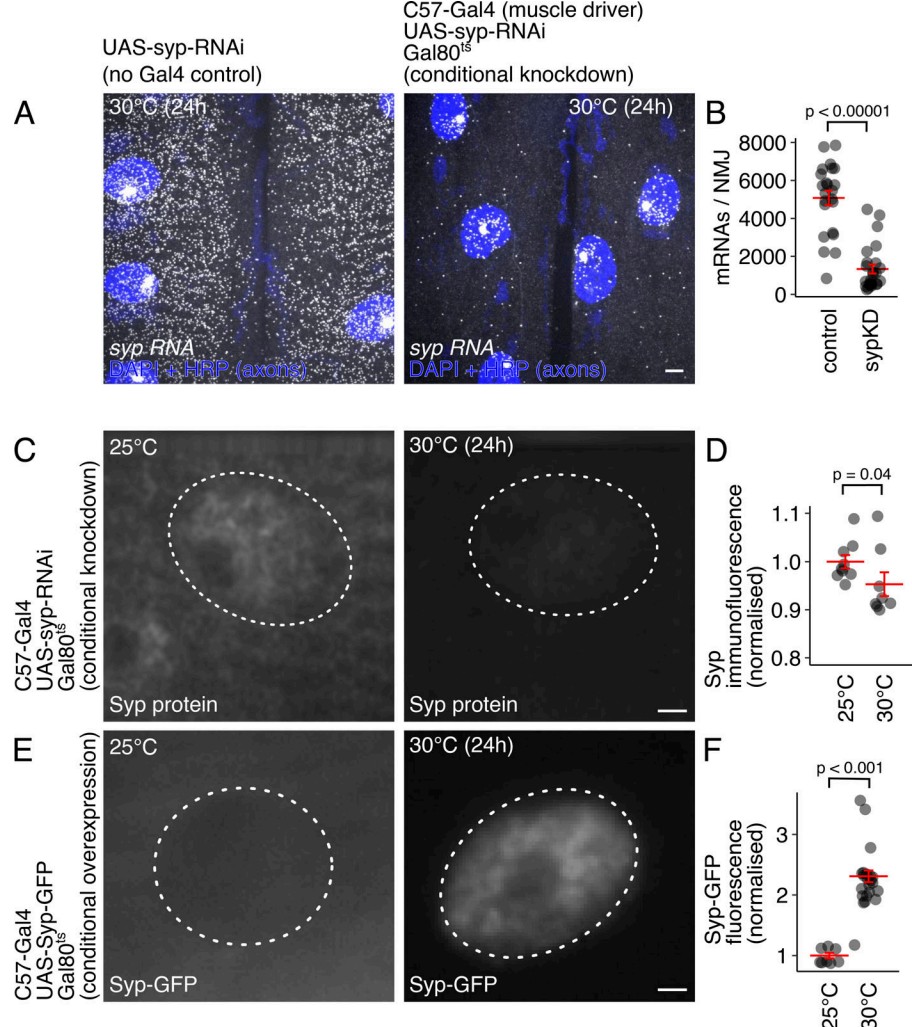

Figure S3. **Quantification of conditional *syp* knockdown and overexpression. (A)** *syp* mRNA levels are dramatically decreased by conditional knockdown, i.e., transferring third instar larvae (tub-Gal80$^{ts}$; C57-Gal4>Syp-RNAi) to the restrictive temperature (30°C) for 24 h. Maximum z-projections of spinning disk confocal images showing *syp* mRNA detection with smFISH. **(B)** Quantification of smFISH images from control and *syp* knockdown NMJs show a significant decrease in *syp* mRNA levels (mean ± SEM; Student's *t* test; *n* = 5 NMJs/condition). **(C)** Immunofluorescence images show that Syp protein levels are conditionally reduced by expressing *syp* RNAi specifically during third larval instar stage. **(D)** Quantification of average Syp protein levels shows a significant reduction in Syp protein grown at the Gal 80 restrictive temperature (mean ± SEM; Student's *t* test; *n* = 10 NMJs/condition, average of three nuclei/NMJ). **(E)** Immunofluorescence images showing that Syp-GFP is conditionally overexpressed in muscle by shifting larvae to the restrictive temperature. **(F)** Quantification of Syp-GFP fluorescence shows that protein levels are significantly elevated at the Gal80 restrictive temperature (mean ± SEM; Student's *t* test; *n* = 10 NMJs/condition, average of three nuclei/NMJ; scale bar = 2 µm).

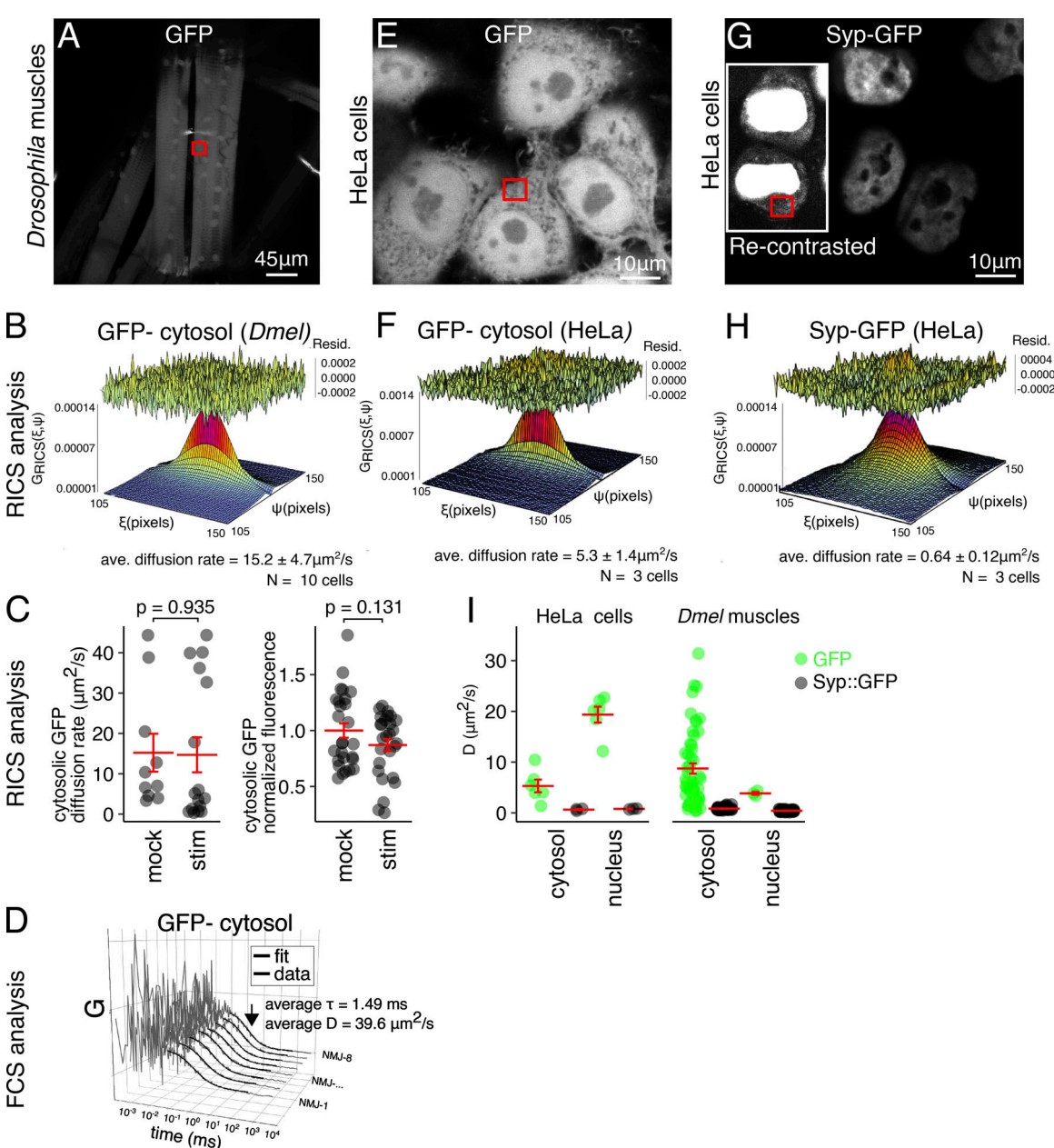

Figure S4. **Syp-GFP diffusion is >10 times slower than GFP in *Drosophila* muscle and human cells. (A)** GFP expression in live larval muscle cells, maximum z-projection of confocal image. Red box indicates the region of interest (ROI) imaged for RICS data acquisition. **(B)** Representative fit of the autocorrelation function from the ROI in A, used to estimate the rate of diffusion. ave, average. **(C)** Cytosolic GFP diffusion and fluorescence intensity in larval muscles are unaffected by KCl stimulation. **(D)** Representative fluorescence correlation spectroscopy (FCS) curves acquired from the ROI in A as an independent measure of diffusion rate. The curves were fitted using a standard 3D, one-species diffusion model and showed diffusion rates similar to RICS measurements. **(E)** Live, confocal image of doxycycline-induced GFP expression in HeLa cells. **(F)** Representative fit of the autocorrelation function from the ROI labeled in E. **(G)** Live, confocal image of doxycycline-induced human Syp-GFP expression in HeLa cells. Inset shows the same image recontrasted to reveal cytosolic Syp expression, which is much lower than nuclear expression. **(H)** Representative fit of the autocorrelation function from the ROI labeled in G. **(I)** Comparison of average diffusion rates for GFP and Syp-GFP in different compartments of *Drosophila* muscle cells and HeLa cell types.

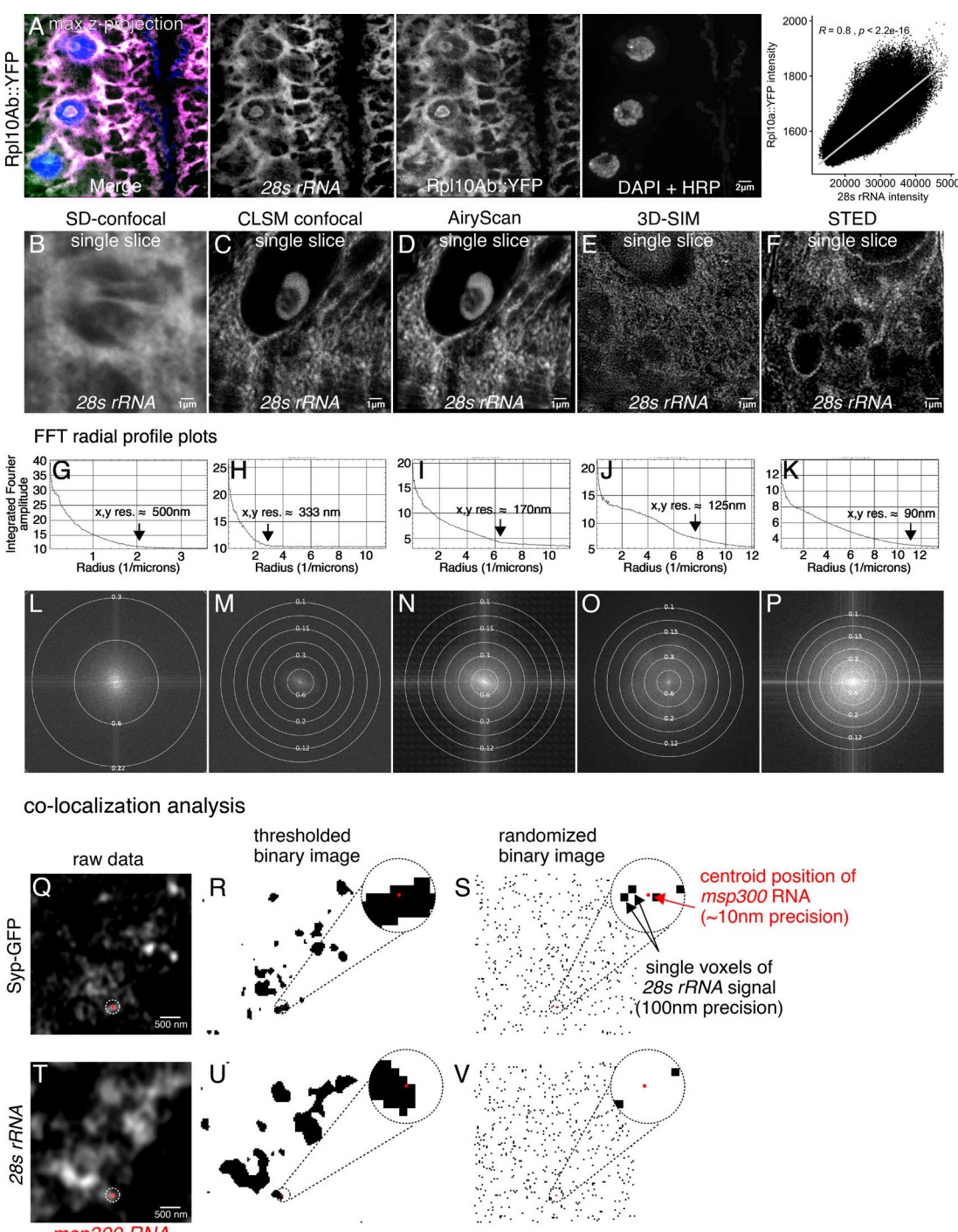

Figure S5. **Detection of ribosome clusters using an smFISH probe targeting 28s rRNA. (A)** Signal from the 28s rRNA smFISH probe overlaps almost completely with signal from Rpl10Ab::YFP protein trap in larval muscles. Images are maximum z-projections from a spinning disk confocal. Quantification of individual pixel intensity for 28s rRNA and Rpl10Ab::YFP signal shows highly significant correlation (Pearson correlation). **(B–F)** AiryScan confocal provides adequate resolution of ribosome clusters in the larval NMJ. Comparison of different confocal and superresolution microscopy techniques for imaging ribosome clusters in the NMJ with the 28s rRNA smFISH probe. The confocal LSM confocal image (C) is from a single AiryScan detector of the same image that has been processed in D. The large pinhole (~3.5 a.u.) makes for an image with suboptimal resolution relative to a properly acquired LSM image. **(E)** It is not possible to achieve the highest resolution enhancement with 3D-SIM due to high background signal that interferes with stripe contrast from the projected SIM pattern. **(F)** With STED, we were able to achieve lateral resolution <100 nm, revealing small clusters of ribosomes within the nuclear envelope and at the postsynaptic density. Of the techniques tested, STED provided the best resolution; however, excessive photon damage from the STED laser prohibited 3D sectioning. Therefore, we chose to perform colocalization experiments using the AiryScan system, which still enabled visualization of ribosome clusters that were observed in STED. **(G–P)** Fast Fourier transform radial profile plots provide a quantitative estimate of the resolution of each system. **(Q–V)** Quantification of msp300 colocalization with Syp-GFP and 28s rRNA. Centroid positions of msp300 punctae were mapped onto thresholded binary images of Syp-GFP (R) and 28s rRNA signal (U). The percentage of colocalized molecules was then compared with a randomized distribution of the same number of pixels (S and V).

JCB

**Provided online is one table in Excel. Table S1 shows individual probe sequences used for smFISH.**

