## [Peer Review File · The Journal of Cell Biology]

Syncrip/hnRNPQ is required for activity-induced Msp300/Nesprin1 expression and new synapse formation

Joshua Titlow, francesca Robertson, Aino Jarvelin, David Ish-Horowicz, Carlas Smith, Enrico Gratton, and Ilan Davis

Corresponding Author(s): Ilan Davis, University of Oxford

Review Timeline:

Submission Date:	2019-03-22
Editorial Decision:	2019-05-08
Revision Received:	2019-08-21
Editorial Decision:	2019-09-20
Revision Received:	2019-11-20
Editorial Decision:	2019-11-26
Revision Received:	2019-12-09

Monitoring Editor: Jodi Nunnari

Scientific Editor: Marie Anne O'Donnell

Transaction Report:

DOI: <https://doi.org/10.1083/jcb.201903135>

May 8, 2019

Re: JCB manuscript #201903135

Prof. Ilan Davis
University of Oxford
Department of Biochemistry
South Parks Road
Oxford OX1 3QU
United Kingdom

Dear Prof. Davis,

Thank you for submitting your manuscript entitled "Syncrip/hnRNPQ is required for activity-induced Msp300/Nesprin1 expression and new synapse formation". The manuscript was assessed by expert reviewers, whose comments are appended to this letter. We invite you to submit a revision if you can address the reviewers' key concerns, as outlined here.

You will see that the reviewers are interested in the proposal that Syncrip-dependent transport of Msp300 is necessary for local translation of Msp300 and synapse remodeling but recommend additional experimental work to more substantively support the study's main claims. We consider all of the points reasonable and feasible to address for resubmission but it is particularly critical to bolster the claim that local translation of Msp300 is occurring and necessary for it to accumulate at new boutons; provide more convincing evidence that Syncrip and Msp300 colocalisation is occurring and required; and that Syncrip-mediated transport of Msp300 is necessary for remodeling in response to a more physiological stimulus. Rev#1 also recommends testing whether boutons are forming in the mutants but just not maturing.

GENERAL GUIDELINES:

Text limits: Character count for an Article is < 40,000, not including spaces. Count includes title page, abstract, introduction, results, discussion, acknowledgments, and figure legends. Count does not include materials and methods, references, tables, or supplemental legends.

Figures: Articles may have up to 10 main text figures. Figures must be prepared according to the policies outlined in our Instructions to Authors, under Data Presentation, <http://jcb.rupress.org/site/misc/ifora.xhtml>. All figures in accepted manuscripts will be screened prior to publication.

Supplemental information: There are strict limits on the allowable amount of supplemental data. Articles may have up to 5 supplemental figures. Up to 10 supplemental videos or flash animations are allowed. A summary of all supplemental material should appear at the end of the Materials and methods section.

The typical timeframe for revisions is three months; if submitted within this timeframe, novelty will not be reassessed at the final decision. Please note that papers are generally considered through only one revision cycle, so any revised manuscript will likely be either accepted or rejected.

Thank you for this interesting contribution to Journal of Cell Biology. You can contact us at the journal office with any questions, cellbio@rockefeller.edu or call (212) 327-8588.

Sincerely,

Jodi Nunnari, Ph.D.
Editor-in-Chief

Marie Anne O'Donnell, Ph.D.
Scientific Editor

Journal of Cell Biology

Reviewer #1 (Comments to the Authors (Required)):

The manuscript by Titlow and colleagues builds on the group's prior findings that RNA-binding protein Syncrip binds msp300 mRNA to regulate its translation. Msp300 is required for new bouton formation at *Drosophila* NMJs in a high-K⁺ stimulation assay. The authors demonstrate that new bouton formation in this assay also depends on Syncrip. Further, animals heterozygous for both syncrip and msp300 fail to form new boutons in response to high-K⁺ stimulation, consistent with a close functional relationship. Through *in vivo* imaging experiments, the authors argue that postsynaptic Syncrip is upregulated upon stimulation and colocalizes with msp300 mRNA and ribosomes in ribonucleoprotein particles (RNPs) to promote translation of new MSP300.

Specific points to address:

1. The authors never demonstrate that translation of msp300 is required for its postsynaptic accumulation at new boutons. This is a critical experiment that needs to be done to support the main conclusion of the manuscript.
2. In Figure 1, the authors should include negative controls showing Msp300 immunofluorescence in msp300 nulls for comparison.

3. In Figure 2, why isn't postsynaptic expression of Msp300 around ghost boutons analyzed? As shown in Figure S1 and in the model in Figure 8, this seems to be the relevant pool of Msp300 protein.

4. In Figure 5, the authors should provide additional evidence that the colocalization they are measuring is biologically significant. A more compelling negative control than free GFP would be a comparison to colocalization with an mRNA that is present in postsynaptic compartments but not regulated by activity. RP49 was used as a negative control in 5A and could perhaps serve as a negative control for the colocalization experiments as well. For comparison, it would also be useful to know how the colocalization of msp300 mRNA and Syncrip compares with levels of colocalization observed when their patterns are randomized.

5. Figures 6 and 7 are swapped.

6. In Figure S4, free GFP is not the ideal negative control. It would be more convincing if the authors looked at an endogenous postsynaptic protein that is not targeted by activity or involved in translation.

7. What percent of msp300 mRNA puncta colocalize with Syncrip and 28s rRNA? Again, it would be useful to compare this to their coincidence when localization of one or more components is randomized.

8. Demonstrating that a second, more physiological, stimulus has a similar effect would support the biological relevance of these findings.

9. Have the authors looked at ghost bouton formation in syncrip, msp300 double mutants for pathway analysis?

10. The authors should note what NMJ they are analyzing and, along with numbers for synapses analyzed, provide data on the number of larvae analyzed.

Reviewer #2 (Comments to the Authors (Required)):

In this manuscript, authors provided a series of biochemical, biophysical and genetic evidences showing that activity-dependent Syp-containing ribosomal granule regulates new synapse formation. By using *Drosophila* neuromuscular junction, authors found Syp is a positive regulator of Msp300 translation, and indeed Msp300 mRNA and Syp make a molecular complex in vivo. Author used genetic and electrophysiological approach to show that Syp is required for activity-dependent plasticity during NMJ development. Authors elegantly took advantage of Gal80[ts] system to avoid the developmental role of Syp. Furthermore, authors used ccRICS and RICS, sophisticated molecular imaging techniques to measure protein colocalization, amount and diffusion rate in vivo and in vitro. Overall, the experimental strategy is very well designed and obtained results are convincing. This manuscript provides invaluable information to understand the molecular principle of neural plasticity. I am convinced with currently provided data and explanations. In sum, the quality of this manuscript is high enough and contains sufficient general interest to justify the publication in *The Journal of Cell Biology*. I suggest authors to address following minor points before the publication.

Minor comments:

1, The detailed molecular structure of Msp300-YFP should be explained in the main text. It is important if 3'-UTR is retained and how YFP is connected.

2, Be careful about Figure 6 and 7. In several sentences (Page 7-11), Figure 6 and 7 are incorrectly described. Or, the order of figures is incorrect.

Reviewer #3 (Comments to the Authors (Required)):

The paper by Davies and colleagues examines the relationship between the RNA-binding protein Syncrip and one of its targets, msp300 at the *Drosophila* neuromuscular junction. In a previous paper the authors show with a pull-down assay that Syncrip and msp300 interact. They examine the interactions between Syncrip and msp300 using imaging. The authors also investigate the possibility that msp300 is translated in an activity-dependent manner.

I have the following questions/critiques/suggestions:

1. In Figure 1 the authors examine the effect of Syncrip loss on Msp300 mRNA and protein. To analyze the mRNA they count the number of Msp300 particles. In the images shown, while the particle number is apparently not affected by Syncrip loss, the intensity appears to be. This is apparent by an overall reduced intensity of the mRNA signal in Figure 1A (compare top and bottom right panels) and also by a reduced intensity of the Msp particles present in the nucleus in the Syp mutant. This should be analyzed and addressed.

2. Why is the axonal morphology so different between the wild-type and Syp mutant images in Figure 1C.?

3. In figure 2 the authors examine the Syp-dependence of Msp300 protein expression in response to "elevated activity". More information on the nature and duration of the elevated activity should be provided in the main text. In the Methods it states that 5 applications of high K⁺ (what concentration?) were applied for 15 min, separated by 5 min and then after 150 min the fly was fixed and processed. (What then is "spaced" application of K⁺ described in the figure legend?) Given the long duration of the experiment (230 min!) how then can one conclude that Msp300 is locally translated? Msp300 translated elsewhere could be targeted to the analyzed location.

3. In figure 2, it is clear that the authors are examining Msp300-YFP protein, not endogenous Msp300. I could not find a single description of this construct. Is it a knock-in? If not a knock-in, how does one interpret the high levels of MSP300 mRNA (endogenous + overexpressed) available for binding to Syncrip? The total amount of MSP300 mRNA expressed should be measured.

4. Did the construct contain the endogenous 5' and 3'UTRs of Msp300? Are the binding sites of Syp known? The Msp300-YFP image shown is blurry and should be improved.

5. In figure 3, the authors examine the effects of KCl stimulation on the mEPSP frequency. There are two missing features to this analysis: first, the baseline level of mEPSPs (before KCl) and second an analysis of mEPSP amplitude, (baseline, after KCl and +/- syp mutation), which is also clearly affected.

Our responses to Reviewer #1's comments

1. “The authors never demonstrate that translation of *msp300* is required for its postsynaptic accumulation at new boutons. This is a critical experiment that needs to be done to support the main conclusion of the manuscript.”

To address this comment, we have performed an additional experiment to determine if *msp300* RNA is translated locally, in proximity to new synapses. In our first submission, we showed that *msp300* was co-localized with Syp and ribosomes at the larval NMJ, which is consistent with local translation. We have now performed a new set of experiments to strengthen this conclusion, in which we show that *msp300* mRNA is present in sub-synaptic granules containing the essential translation initiation factor eIF4E, which has previously been shown to be required for post-synaptic translation of GluRIIA at the NMJ synapse (Sigrist et al., 2000- *Nature*). Crucially, we show that the number of *msp300* and eIF4E containing granules doubles in response to elevated synaptic activity, strongly suggesting that translation of *msp300* increases at the synapse in response to activation. Our results are consistent with a number of long standing studies showing that eIF4E is a rate limiting factor in translation initiation (Lazaris-Karatz et al., 1990- *Nature*) and accumulates in large granules along with poly-A binding protein at the larval NMJ during experience-dependent synaptic growth (Sigrist et al., 2000- *Nature*; Sigrist et al., 2002- *J Neurosci*). **We have revised the manuscript to add these new results to page 10, and in Figure 7.**

2. “In Figure 1, the authors should include negative controls showing Msp300 immunofluorescence in *msp300* nulls for comparison.”

Good point. We have added the negative controls for Msp300 immunofluorescence to **Fig. S2E**, showing that there is no signal when Msp300 protein is absent.

3. “In Figure 2, why isn't postsynaptic expression of Msp300 around ghost boutons analyzed? As shown in Figure S1 and in the model in Figure 8, this seems to be the relevant pool of Msp300 protein.”

This is a good point. The reason for not measuring the pool of Msp300 protein directly around ghost boutons is that ghost boutons are very rarely observed in the *syp* knock down lines, which prevents us from comparing accumulation across conditions or genotypes at the ghost boutons. Furthermore, the model based on our data is that Msp300 is produced close to the synapses rather than the exact site where the protein becomes enriched, which is still local to the synapses, considering that the muscle nuclei are much further away. Therefore, we have analyzed Msp300 expression within a few microns of the NMJ, which is representative of the 'local' newly

synthesized pool of Msp300 that contributes to enrichment around ghost boutons, and ultimately functions to stabilize new synapses. **To address this point we have revised the manuscript to include an explanation in the Discussion of what we mean by “local”** in the context of the NMJ. We have also **revised our model in Fig. 8** to clarify that ‘local’ translation is not restricted to the ghost boutons *per se*. **In addition, we have revised the schematic and figure legend in Fig. 2A** to emphasize that the region analyzed was near the synapse, not the synapse itself.

4. “In Figure 5, the authors should provide additional evidence that the colocalization they are measuring is biologically significant. A more compelling negative control than free GFP would be a comparison to colocalization with an mRNA that is present in postsynaptic compartments but not regulated by activity. RP49 was used as a negative control in 5A and could perhaps serve as a negative control for the colocalization experiments as well.”

To address this comment, we show in a control experiment that Syp binds to a non-specific *RNA* (*rp132*, current name for RP49) significantly less than to *msp300* mRNA. This experiment involved micro-injecting Cy5-labelled *rp132* into muscle cells expressing Syp-GFP and performing ccRICS analysis, **as shown in Figure 5**. Our data analysis showed that the proportion of *rp132* molecules interacting with Syp is several orders of magnitude lower than the proportion observed for *msp300*.

“For comparison, it would also be useful to know how the colocalization of msp300 mRNA and Syncrip compares with levels of colocalization observed when their patterns are randomized.”

To address this comment, we have now added a comparison of the quantification of *msp300* and Syp co-localisation after randomising pixel distributions and testing for statistical significance (**Fig. 7G,N, and U**). In each case co-localization between *msp300* and other molecules in the complex (Syp, *28s rRNA*, or both) is much higher than co-localization with randomized voxels (t-test, $p < 0.0001$). Therefore, we conclude that the observed associations are biologically relevant. **Details of the analysis are provided in Materials and Methods section- Co-localization analysis.** Note that it is not possible to perform similar randomization tests for ccRICS data because there is no autocorrelation from randomized signal. Without an autocorrelation function, it is not possible to obtain the requisite parameters from the ccRICS fitting procedure.

5. “Figures 6 and 7 are swapped.”

Thank you. **Now corrected.**

6. “In Figure S4, free GFP is not the ideal negative control. It would be more convincing if the authors looked at an endogenous postsynaptic protein that is not targeted by activity or involved in translation.”

We understand why the reviewer is suggesting such a control, but it is very difficult to identify a protein that is known for certain to not be targeted by activity or involved in translation. The experiment we show in Fig. S4C is a control, using GFP, for the general diffusive properties and translational capacity of the cellular environment. We feel that GFP is not only an excellent control for general diffusion of a protein with no connection with synaptic plasticity, but GFP is also part of the fusion construct used in the experiment. Therefore, our results show convincingly that the decreased mobility and increased intensity of Syp in response to activation (Fig. C and F) are specific to Syp, and not a consequence of a general change in the diffusion of proteins in the cytoplasm, nor the effect of the GFP moiety of the Syp-GFP fusion.

7. “What percent of *msp300* mRNA puncta colocalize with Syncrip and 28s rRNA? Again, it would be useful to compare this to their coincidence when localization of one or more components is randomized.”

To address this comment, we have quantified co-localization between *msp300* mRNA and compared it to randomized Syp and 28s rRNA signal. We found that 11.2% of *msp300* molecules co-localize with both Syp and 28s rRNA, whereas only 1.9% of *msp300* molecules co-localize when Syp and 28s rRNA are randomized. We obtained similar results when only individual components were randomized. We have revised the manuscript to include this new data (**Fig 7**).

8. “Demonstrating that a second, more physiological, stimulus has a similar effect would support the biological relevance of these findings.”

This is an excellent suggestion. To address this point we visualized Msp300 protein in the larval NMJ after electrical stimulation. We observed similar rapid enrichment of Msp300 at GBs that has been observed by KCl stimulation (**Figure. S1B**). Therefore, we conclude that the molecular phenomena observed using KCl stimulation in this study are biologically relevant. These data are in good agreement with the literature. It has previously been shown that patterned stimulus by KCl is very comparable to patterned electrical or optogenetic activation of the motor axon with regard to ghost bouton (GB) formation and potentiation of mEPSP frequency (Ataman et al., 2008). We have added the following text to clarify this point:

“To determine how activity-dependent enrichment of Msp300 at the synapse is regulated we measured the effect of Syp on Msp300 protein and mRNA levels in KCl stimulated

samples, a well characterized model for inducing synaptic plasticity that mimics behavioral, electrical, and optogenetic induction of synapse growth in the larval NMJ (Ataman et al., 2008; Sigrist et al., 2003)."

We have revised the manuscript to include this new data (Figure S1B, Page 30).

9. "Have the authors looked at ghost bouton formation in syncrip, msp300 double mutants for pathway analysis?"

We have not looked at *syp; msp300* double mutants. Both *syp* and *msp300* mutants are quite sick (neither make it to adults), so we expect the double mutant larvae to be even sicker. Moreover, given the strong NMJ developmental phenotypes that have been reported for either mutant alone and the strong synaptic plasticity phenotypes we observe in trans-heterozygotes (Fig. 4), we feel that such double mutants would add little to this study.

10. "The authors should note what NMJ they are analyzing and, along with numbers for synapses analyzed, provide data on the number of larvae analyzed."

Thank you for those suggestions. We have added the following text to the **Materials and Methods** section to be more transparent about the issues raised by the reviewer:

"NMJs at muscles 6 and 7 in segments 3-5 were imaged for at least five different larvae per condition/genotype and multiple cells per larvae unless specified otherwise."

Our responses to Reviewer #2's comments

1. "The detailed molecular structure of Msp300-YFP should be explained in the main text. It is important if 3'-UTR is retained and how YFP is connected."

Thank you for this excellent suggestion. These flies are from a previously published knock-in line that was generated by the Cambridge Protein Trap Insertion Consortium (Lowe et al, 2014-*Development*). Therefore, the *msp300::yfp* transcripts contain the endogenous 3'-UTR. We have now included information about the specific line that was used in the *Drosophila* maintenance subsection of **Materials and Methods Section**.

2. "Be careful about Figure 6 and 7. In several sentences (Page 7-11), Figure 6 and 7 are incorrectly described. Or, the order of figures is incorrect."

Thank you for pointing out this error. **We have now switched the figures to the correct order.**

Our responses to Reviewer #3's comments

1. “In Figure 1 the authors examine the effect of Syncrip loss on Msp300 mRNA and protein. To analyze the mRNA they count the number of Msp300 particles. In the images shown, while the particle number is apparently not affected by Syncrip loss, the intensity appears to be. This is apparent by an overall reduced intensity of the mRNA signal in Figure 1A (compare top and bottom right panels) and also by a reduced intensity of the Msp particles present in the nucleus in the Syp mutant. This should be analyzed and addressed.”

It is true that the representative images in Figure 1A show that transcription (number of primary transcripts) is reduced in the mutant (lower panel). Our quantification in Figure 3C shows that transcription (number of primary transcripts / nucleus) is not affected across the whole dataset. The representative images were chosen to convey the average number of cytosolic transcripts. We have replaced those images with images that convey the equivalent number of mature and primary transcripts, which are more representative of the data.

Regarding the reviewer's comment about overall reduced intensity in the mRNA signal, we do not observe differences in relative smFISH signal intensity between genotypes. An advantage of smFISH is that it quantitates RNA content in absolute numbers from counting individual foci representing single mRNA molecules, as opposed to relative intensity. We don't observe differences in the intensity of individual foci, so the number of foci provide an absolute measure of number of mRNA molecules.

2. “Why is the axonal morphology so different between the wild-type and Syp mutant images in Figure 1C.?”

The reason for the difference in axonal morphology which the reviewer correctly points out is that *syp* affects NMJ development, a phenotype we have previously reported in detail (McDermott et al., 2014). That is why we also performed experiments where Syp expression was conditionally knocked down specifically in the larval stage using the Gal80 system (Figure 3D). **We have added some text in the Results section to clarify the *syp* NMJ phenotype and our motivation for performing that experiment:**

“Importantly, the NMJ morphology in conditional syp knockdown mutants was indistinguishable from wild type NMJs (Fig. 3D), indicating that synapse development was normal (as opposed to the NMJ axon developmental overgrowth phenotype observed in syp^{e00286}, Fig. 1C and (McDermott et al., 2014)).”

3. “In figure 2 the authors examine the Syp-dependence of Msp300 protein expression in response to “elevated activity”. More information on the nature and duration of the elevated activity should be provided in the main text. In the Methods it states that 5 applications of high K+ (what concentration?) were applied for 15 min, separated by 5 min and then after 150 min the fly was fixed and processed.”

We have now added the following details to the methods section, rather than expecting the reader to look up the details in Altman et al. 2008: KCl concentration was 90mM (see next comment too).

“What then is “spaced” application of K+ described in the figure legend?”

Flies were fixed 150 minutes after the first stimulus, that includes the spaced stimulations. KCl pulses were 2-6mins (2, 2, 2, 4, and 6mins respectively, following very precisely the protocol from Ataman et al., 2008- *Neuron*). **We have modified the text to include these details:**

“A series of 5 short high potassium saline (KCl, 90mM) pulses (2, 2, 2, 4, and 6mins respectively) were separated by 15min perfusion of HL3 precisely as described previously (Attaman et al., 2008).”

“Given the long duration of the experiment (230 min!) how then can one conclude that Msp300 is locally translated? Msp300 translated elsewhere could be targeted to the analyzed location.”

The total length of the experiment is 150', not 230', but still the reviewer is correct. It is possible that protein could be translated elsewhere as well as near the synapse. In the discussion section we had already addressed this possibility, but we have now added an additional paragraph in the Discussion section (Page 12) to hopefully make that point even stronger:

“Local’ translation in neurons refers to protein synthesis that occurs within axons or dendrites, independent of the cell body, and usually in response to a specific stimulus that induces synaptic plasticity. The distance between the site of translation and the functional

site of the new proteins varies from 1-20 μ m depending on cell type (Rangaraju et al., 2017)....”

3. "In figure 2, it is clear that the authors are examining Msp300-YFP protein, not endogenous Msp300. I could not find a single description of this construct. Is it a knock-in? If not a knock-in, how does one interpret the high levels of MSP300 mRNA (endogenous + overexpressed) available for binding to Syncrin? The total amount of MSP300 mRNA expressed should be measured."

Thank you for pointing out that we did not include information about the Msp300::YFP flies. These flies are knock-ins that were generated by the Cambridge Protein Trap Insertion Consortium (Lowe et al., 2014- *Development*). We have corrected this omission by including information about the specific line that was used **in the Materials and Methods Section**.

4. "Did the construct contain the endogenous 5' and 3'UTRs of Msp300?"

Endogenous UTRs are retained in the Msp300::YFP larva, as the YFP coding sequence inserts into an intron of the endogenous locus. We have made this clearer in the relevant **Methods section and Figure S2**, which shows a map of all of the *msp300* isoforms and the relative position of the YFP insert.

"Are the binding sites of Syp known?"

Syp binding sites on *msp300* are not known. We are currently in the process of obtaining CLIP data for Syp in *Drosophila*, but this is a totally separate project that falls outside the scope of the current work.

"The Msp300-YFP image shown is blurry and should be improved."

The Msp300::YFP images in Fig. 2B are raw data. We agree that they are not the most aesthetic and have now processed them with a basic deconvolution algorithm (Richardson-Lucy, 20 iterations using the DeconvolutionLab2 plugin in ImageJ and a synthetic PSF (Airy function)). This algorithm does not affect the total intensity of the image, only the contrast for display purposes, therefore quantification of the dataset is unchanged. **The following text was added to the Materials and Methods section:**

“Images in Fig. 2B were deconvolved for display purposes using the Richardson-Lucy algorithm in the ImageJ DeconvolutionLab2 plugin (20 iterations, Airy PSF (Sage et al., 2017).”

5. “In figure 3, the authors examine the effects of KCl stimulation on the mEPSP frequency. There are two missing features to this analysis: first, the baseline level of mEPSPs (before KCl) and second an analysis of mEPSP amplitude, (baseline, after KCl and +/- syp mutation), which is also clearly affected.”

These are good points. Baseline voltage/time traces were added to **Fig. 3C**. Analysis of mEJP amplitude has also been included in **Fig. 3C**. We did not observe activity-dependent potentiation of mEJP amplitude, or a significant difference between genotypes. Different representative traces were chosen to avoid giving the impression that there were any differences in mEJP amplitude. This is consistent with previous analyses which showed that baseline mEJP frequency and amplitude are not affected by loss of *syp* function (Halstead et al., 2014).

September 20, 2019

Re: JCB manuscript #201903135R

Prof. Ilan Davis
University of Oxford
Department of Biochemistry
South Parks Road
Oxford OX1 3QU
United Kingdom

Dear Prof. Davis,

Thank you for submitting your revised manuscript entitled "Syncrin/hnRNPQ is required for activity-induced Msp300/Nesprin1 expression and new synapse formation". The manuscript has been seen by two of the original reviewers whose full comments are appended below. Overall, the reviewers are impressed with your responses, but there are a few additional/remaining issues that need to be addressed.

First, there is continued concern that association with the translation initiation factor eIF4E is definitive evidence for new translation (expressed by the second reviewer in a note to the editors). Appreciating the difficulties in using modified amino acids or ribosome protection/affinity approaches in this system, I would suggest that you revise aspects of your text to reflect this limitation. Second, the first two comments of the first reviewer seem reasonable to me and I strongly urge that you collect and incorporate this data into your final submission. I do not expect you to perform the experiments recommended in this reviewer's item 3 (double homozygotes). Additionally, there are a number of more minor comments that I ask that you consider. For clarification, I should let you know that the second reviewer expressed general satisfaction with the exception of the eIF4E reservation described above.

I hope you can perform the requested analyses quickly and return the manuscript to the journal. I expect to make a final decision editorially.

Our general policy is that papers are considered through only one revision cycle; however, given that the suggested changes are relatively minor we are open to one additional short round of revision.

Please submit the final revision within one month, along with a cover letter that includes a point by point response to the remaining reviewer comments.

Thank you for this interesting contribution to the Journal of Cell Biology. You can contact me or the scientific editor listed below at the journal office with any questions, cellbio@rockefeller.edu or call (212) 327-8588.

Sincerely yours,

Louis Reichardt, Ph.D.

Monitoring Editor

Marie Anne O'Donnell, Ph.D.
Scientific Editor

Journal of Cell Biology

Reviewer #1 (Comments to the Authors (Required)):

The manuscript by Titlow and colleagues builds on the group's prior findings that RNA-binding protein Syncrip binds msp300 mRNA to regulate its translation. Msp300 is required for new bouton formation at *Drosophila* NMJs in a high-K⁺ stimulation assay. The authors demonstrate that new bouton formation in this assay also depends on Syncrip. Further, animals heterozygous for both syncrip and msp300 fail to form new boutons in response to high-K⁺ stimulation, consistent with a close functional relationship. Through in vivo imaging experiments, the authors argue that postsynaptic Syncrip is upregulated upon stimulation and colocalizes with msp300 mRNA and ribosomes in ribonucleoprotein particles (RNPs) to promote translation of new MSP300. The revised manuscript nicely addresses some points raised in the initial round of review, most significantly through the inclusion of the eIF4E analysis to address the role of local translation in the activity-dependent accumulation of Msp300. However, several concerns were not fully addressed:

1. The explanation that Msp300 protein levels at ghost boutons cannot be measured in *syp* mutants for figures 2B-E is not convincing. Figure 3 shows that ghost boutons still form at a rate of 2-3/NMJ in mutants (vs. 5 in control), so it should be possible to look specifically at postsynaptic enrichment of Msp300 protein. The rest of the response addresses RNA, where measurement at more distal sites makes sense.

2. Regarding the addition of a more biological stimulus in Figure S1, the authors should quantify these data as it's difficult to draw any conclusions from an image of a single ghost bouton induced by electrical stimulation. Specifically, what % of ghost boutons exhibit post-synaptic accumulation of Msp300 with both forms of induction?

For stimulation with 90 mM K⁺, it's too strong to say that this 'recapitulates' more biological induction paradigms. The authors themselves note that electrical stimulation has a much smaller effect and the behavioral manipulations they reference lead to mature bouton growth, not ghost boutons. The language just needs to be toned down a bit.

3. The argument that double mutants will add little to this study is not compelling. While the observed double heterozygous interaction suggests a close functional interaction, double mutant analysis will test whether *syp* and *msp300* act in a linear pathway to promote ghost bouton formation, consistent with the model proposed.

In Figure 4, I suggest using mutant allele/+ as a more standard way of indicating heterozygotes than 'wild type'/mutant allele.

4. I would recommend representing individual data points in the graphs throughout.

5. Figure 1 - I believe muscle Msp300 levels were measured, so the title of the figure should refer to

muscle, not NMJ, expression.

6. I didn't see it stated that Ataman et al., 2008 demonstrated that ghost bouton formation is translation dependent. This seems important to mention.

Reviewer #3 (Comments to the Authors (Required)):

The authors have dealt with my criticisms in a satisfactory manner. I believe, however, that they over-state their response to reviewer 1's criticism that they have not shown msp300 translation. Showing an increased association with eIF4E is not the same thing as showing translation. The wording in the manuscript should be changed/softened to reflect this.

DEPARTMENT OF BIOCHEMISTRY
UNIVERSITY OF OXFORD

Micron
OXFORD

Nano

South Parks Oxford, OX1 3QU

Tel: +44 (0) 1865 613265

Lab: +44 (0) 1865 613271

ilan.davis@bioch.ox.ac.uk

<http://ilandavis.com>

<http://micronoxford.com>

Lab manager: darragh.ennis@bioch.ox.ac.uk

PA: jolanta.parkinson@bioch.ox.ac.uk

November 20, 2019

Dear Jodi and Marie:

Please find submitted our revised manuscript 201903135R2 along with our response to yours and the reviewers' comments below. As in the previous round of revisions, we thank the reviewers for providing very useful and insightful suggestions that further improve the manuscript, as well as the very useful guidance of the editors. We have performed two additional experiments and modified the text of the manuscript to directly address each comment, as explained below.

Response to Editors' comments:

1. "First, there is continued concern that association with the translation initiation factor eIF4E is definitive evidence for new translation... I would suggest that you revise aspects of your text to reflect this limitation."

We agree that association with eIF4E is not definitive evidence for new translation and have revised the manuscript to avoid overstating this interpretation. In particular, we have modified the abstract to remove terminology that directly implicates Syp in translation. In the main text we have avoided definitive language when describing the conclusions relating to translation. We also make sure that the discussion section is appropriate in this respect.

2. "Second, the first two comments of the first reviewer seem reasonable to me and I strongly urge that you collect and incorporate this data into your final submission."

We have taken this on board and acquired new data to address the first two comments from Reviewer 1 and provide a detailed description of the results in response to the comments below.

In summary, we quantified Msp300 enrichment at ghost boutons both in *syp* mutants and at electrically stimulated synapses. The latter experiment shows that ghost boutons form in response to a more biologically specific stimulus (individual motor neuron electrical activation; Fig. S1E-G) and that the percentage of GBs with Msp300 enrichment is indistinguishable from KCl-induced GB formation (Fig. S1H). The former experiment shows that Msp300 enrichment at ghost bouton occurs in the few ghost boutons that do form in *syp* mutants (far fewer than wild type). We interpret this finding to mean that activity-dependent synaptic enrichment of Msp300 can come from other sources, but that these sources alone are not sufficient for normal levels of GB formation. We also suggest that *msp300* is probably not the only mRNA that requires Syp regulation during synaptic plasticity.

Response to Reviewer #1's comments:

1. "The explanation that Msp300 protein levels at ghost boutons cannot be measured in *syp* mutants for figures 2B-E is not convincing. Figure 3 shows that ghost boutons still form at a rate of 2-3/NMJ in mutants (vs. 5 in control), so it should be possible to look specifically at postsynaptic enrichment of Msp300 protein. The rest of the response addresses RNA, where measurement at more distal sites makes sense."

This is a good point. To address it, we have acquired new data to measure Msp300 enrichment at ghost boutons in *syp* mutants compared to the *syp* rescue line (Fig. S1I-P). We find that Msp300 levels and the percentage of ghost boutons with Msp300 enrichment are not affected by loss of *syp*. This result also indicates that Msp300 accumulation at the synapse can arise from other sources, but that the process is not sufficient for proper activity-dependent bouton growth. Our findings suggest that other *Syp* targets are important for activity-dependent synapse formation. We have edited the Results and Discussion sections to incorporate this additional experiment. We thank the reviewers for this comment, as we feel it has certainly improved the manuscript.

The following text has been added to report the result:

pg. 5: “Surprisingly, the few GBs that formed in the absence of *Syp* showed wild type levels of Msp300 enrichment (Fig. S1I-P).”

The following statements have been added to explain the data more precisely:

pg. 3: “... we show that *Syp* is required for ... regulating activity-induced Msp300 expression ~~at the NMJ~~ in larval muscles.”

pg. 11: “We observe Msp300 at GBs in the absence of *syp* (Fig. S1I-P), however these alternative sources of Msp300 are not sufficient for producing normal levels of activity-dependent bouton growth (Fig. 3B,E). It is also likely that *Syp* is required for the activity-dependent accumulation of other proteins required for synapse formation, as *Syp* is known to bind dozens of mRNAs coding for synaptic proteins (McDermott et al., 2014).”

2. “Regarding the addition of a more biological stimulus in Figure S1, the authors should quantify these data as it's difficult to draw any conclusions from an image of a single ghost bouton induced by electrical stimulation. Specifically, what % of ghost boutons exhibit post-synaptic accumulation of Msp300 with both forms of induction?”

This is also a very good point. To address it, we have acquired additional data to quantify the percentage of ghost boutons that accumulate Msp300 in response to 40Hz motor nerve stimulation (Fig. S1A-H; 46 NMJs from a total of 6 larvae). We observed as many as 6 ghost boutons per NMJ in response to electrical stimulation (Fig. S1D-F), and although on average there were fewer than 1 ghost bouton per NMJ (Fig. S1G), ~90% of the NMJs had Msp300 enrichment, which was statistically indistinguishable from KCl-stimulated NMJs. Therefore, we conclude that KCl activation, which has been widely used in the NMJ and other nervous system preparations (such as mouse brain sections and neuronal cell culture systems), is a biologically meaningful synaptic plasticity assay specifically with respect to Msp300.

“For stimulation with 90 mM K+, it's too strong to say that this 'recapitulates' more biological induction paradigms. The authors themselves note that electrical stimulation has a much smaller effect and the behavioral manipulations they reference lead to mature bouton growth, not ghost boutons. The language just needs to be toned down a bit.

We have taken this view on board, and have toned down the description of the KCl assay as follows:

pg. 4: “... a well characterized model for synaptic plasticity that induces several of the physiological responses associated with behavioral, electrical and optogenetic stimulated synapse growth at the larval NMJ.”

3. “The argument that double mutants will add little to this study is not compelling. While the observed double heterozygous interaction suggests a close functional interaction, double mutant analysis will test whether *syp* and *msp300* act in a linear pathway to promote ghost bouton formation, consistent with the model proposed.”

While it is true that double mutant analysis could provide additional genetic evidence in principle, we have not pursued this experiment because:

- 1) Homozygous *syp* and *mzp300* mutations have a strong developmental phenotype (NMJ is structurally and functionally aberrant and animals do not make it to adulthood), making it difficult to disentangle the developmental effects of genetic suppression or enhancement from genetic interactions that occur in the context of synaptic plasticity.
- 2) In contrast, in the *mzp300/+ ; syp/+* trans-heterozygous larvae the NMJ develops normally, enabling us to assess the interaction of *syp* and *mzp300* specifically during activity-dependent plasticity and in the absence of any defects in NMJ development.
- 3) The trans-heterozygous larval experiment already shows a clear synthetic genetic interaction that is evidence for a functional interaction.
- 4) Our biophysical and biochemical data provide evidence for a physical interaction between Syp protein and *mzp300* mRNA, which is arguably more informative than additional genetic evidence.

To explain this point more clearly, we have modified text in the Results section as follows:

pg.6: "Homozygous syp and mzp300 mutants exhibit strong developmental phenotypes, as both mutants have structurally and functionally aberrant NMJs and the animals do not survive to the adult stage. Therefore, to test for genetic interactions in the context of normally developed larval NMJ synapses we performed a trans-heterozygous genetic interaction experiment in syp^{e00286}/mzp300^{A3} larvae."

"In Figure 4, I suggest using mutant allele/+ as a more standard way of indicating heterozygotes than 'wild type'/mutant allele."

To address this comment, we have changed the genotype notation in Figure 4.

4. "I would recommend representing individual data points in the graphs throughout."

In response, we have changed all of the bar charts to show individual data points where appropriate, i.e., except where the data are percentages.

5. "Figure 1 - I believe muscle Msp300 levels were measured, so the title of the figure should refer to muscle, not NMJ, expression."

To address this point, we have revised the title of the legend in Fig. 1 to "Msp300 expression in larval muscle...".

6. "I didn't see it stated that Ataman et al., 2008 demonstrated that ghost bouton formation is translation dependent. This seems important to mention."

Good point. To address this omission, we have added the following citation to introduce the ribosome/eIF4E experiments.

pg. 9: "Activity-dependent synapse formation at the larval NMJ requires translation (Ataman et al., 2008)."

Response to Reviewer #3's comments:

1. "The authors have dealt with my criticisms in a satisfactory manner. I believe, however, that they over-state their response to reviewer 1's criticism that they have not shown mzp300 translation. Showing an increased association with eIF4E is not the same thing as showing translation. The wording in the manuscript should be changed/softened to reflect this."

We agree with the reviewer that association with eIF4e is not an absolute demonstration of translation. We were careful throughout the manuscript to avoid stating that *mzp300* is translated locally, however there were some specific instances where we mentioned "translation" in the abstract, which we have now modified in the following ways:

pg. 1: "Here, we show that ~~translation of mzp300 mRNA~~ activity-dependent accumulation of Msp300 ~~close to the future site of new synapses~~ in the post-synaptic compartment of

the Drosophila larval neuromuscular junction requires is regulated by the conserved RNA binding protein Syncrip/hnRNP Q."

pg. 1: "These results introduce Syncrip as an important early-acting activity-dependent ~~translational~~ regulator of a plasticity gene..."

We look forward to receiving your decision on the re-revised manuscript. Please do not hesitate to contact us if you require any further information.

With best regards and thanks for a very helpful reviewing process,

I .Davis

Ilan Davis, Josh Titlow and David Ish-Horowicz, and also on behalf of all the other authors.

November 26, 2019

RE: JCB Manuscript #201903135RR

Prof. Ilan Davis
University of Oxford
Department of Biochemistry
South Parks Road
Oxford OX1 3QU
United Kingdom

Dear Prof. Davis:

Thank you for submitting your revised manuscript entitled "Syncrip/hnRNPQ is required for activity-induced Msp300/Nesprin1 expression and new synapse formation". We would be happy to publish your paper in JCB pending final revisions necessary to meet our formatting guidelines (see details below).

- Provide main and supplementary text as separate, editable .doc or .docx files
- Provide figures as separate, editable files according to the instructions for authors on JCB's website, paying particular attention to the guidelines for preparing images at sufficient resolution for screening and production
- Provide table as excel file
- Add a paragraph after the Materials and Methods section briefly summarizing the online supplementary materials
- Add scale bar Fig S2E, S3A

A. MANUSCRIPT ORGANIZATION AND FORMATTING:

Full guidelines are available on our Instructions for Authors page, <http://jcb.rupress.org/submission-guidelines#revised>. **Submission of a paper that does not conform to JCB guidelines will delay the acceptance of your manuscript.**

B. FINAL FILES:

- An editable version of the final text (.DOC or .DOCX) is needed for copyediting (no PDFs).
- High-resolution figure and video files: See our detailed guidelines for preparing your production-

ready images, <http://jcb.rupress.org/fig-vid-guidelines>.

Thank you for this interesting contribution, we look forward to publishing your paper in Journal of Cell Biology.

Sincerely,

Louis Reichardt
Monitoring Editor

Marie Anne O'Donnell, Ph.D.
Scientific Editor

Journal of Cell Biology